# Peri-midFormer: Periodic Pyramid Transformer for Time Series Analysis

**Qiang Wu   Gechang Yao   Zhixi Feng**[†]   **Shuyuan Yang**
Key Laboratory of Intelligent Perception and Image Understanding of
Ministry of Education, School of Artificial Intelligence, Xidian University, China
`{wu_qiang, yao_gechang}@stu.xidian.edu.cn`, `{zxfeng, syyang}@xidian.edu.cn`

## Abstract

Time series analysis finds wide applications in fields such as weather forecasting, anomaly detection, and behavior recognition. Previous methods attempted to model temporal variations directly using 1D time series. However, this has been quite challenging due to the discrete nature of data points in time series and the complexity of periodic variation. In terms of periodicity, taking weather and traffic data as an example, there are multi-periodic variations such as yearly, monthly, weekly, and daily, etc. In order to break through the limitations of the previous methods, we decouple the implied complex periodic variations into inclusion and overlap relationships among different level periodic components based on the observation of the multi-periodicity therein and its inclusion relationships. This explicitly represents the naturally occurring pyramid-like properties in time series, where the top level is the original time series and lower levels consist of periodic components with gradually shorter periods, which we call the periodic pyramid. To further extract complex temporal variations, we introduce self-attention mechanism into the periodic pyramid, capturing complex periodic relationships by computing attention between periodic components based on their inclusion, overlap, and adjacency relationships. Our proposed Peri-midFormer demonstrates outstanding performance in five mainstream time series analysis tasks, including short- and long-term forecasting, imputation, classification, and anomaly detection. The code is available at `https://github.com/WuQiangXDU/Peri-midFormer`.

## 1 Introduction

Time series analysis stands as a foundational challenge pivotal across diverse real-world scenarios [1], such as weather forecasting [2], imputation of missing data within offshore wind speed time series [3], anomaly detection for industrial maintenance [4], and classification [5]. Due to its substantial practical utility, time series analysis has garnered considerable interest, leading to the development of a large number of deep learning-based methods for it.

Different from other forms of sequential data, like language or video, time series data is continuously recorded, capturing scalar values at each time point. Furthermore, real-world time series variations often entail complex temporal patterns, where multiple fluctuations (e.g., ascents, descents, fluctuations, etc.) intermingle and intertwine, particularly salient is the presence of various overlapping periodic components in it, rendering the modeling of temporal variations exceptionally challenging.

Deep learning models, known for their powerful non-linear capabilities, capture intricate temporal variations in real-world time series. Recurrent neural networks (RNNs) leverage sequential data, allowing past information to influence future predictions [6; 7]. However, they face challenges with long-term dependencies and computational inefficiency due to their sequential nature. Temporal convolutional neural networks (TCNs) [8; 9] extract variation information but struggle with capturing long-term dependencies. Transformers with attention mechanisms have gained popularity for sequen-

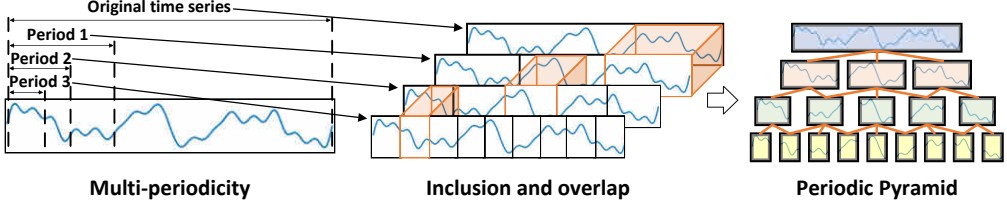

Figure 1: Multi-periodicity, inclusion of periodic components, and Periodic Pyramid.

tial modeling [10; 11], capturing pairwise temporal dependencies among time points. Yet, discerning reliable dependencies directly from scattered time points remains challenging [12]. Timesnet [13] innovatively transforms 1D time series into 2D tensors, unifying intra- and inter-period variations in 2D space. However, it overlooks inclusion relationships between periods of different scales and is constrained by limited feature extraction capability of CNNs, hindering its ability to explore complex relationships within time series.

We analyze time series by examining the inclusion and overlap (hereinafter collectively referred to as inclusion) relationships between various periodic components to address complex temporal variations. Real-world time series often show multiple periodicities, like yearly and daily weather variations or weekly and daily traffic fluctuations. And these periods exhibit clear inclusion relationships, for instance, yearly weather variations encompass multiple daily weather variations. Besides variations between different period levels, it also occur within periods of the same level. For example, daily weather variations differ based on conditions like sunny or cloudy. Due to these inclusion and adjacency relationships, different periods show similarities, with short and long periods being consistent in overlapping portions, and periods of the same level being similar. Additionally, a long period can be decomposed into multiple short ones, forming a hierarchical pyramid structure. In our study, time series without explicit periodicity are treated as having infinitely long periods.

Based on the above analysis, we decompose the time series into multiple periodic components, forming a pyramid structure where longer components encompass shorter ones, termed the Periodic Pyramid as shown in Figure 1, which illustrates the intricate periodic inclusion relationships within the time series. Each level consists of components with the same period, exhibiting the same phase, while different levels contain components with inclusion relationships. This transformation converts the original 1D time series into a 2D representation, explicitly showing the implicit multi-period relationships. Within the Periodic Pyramid, a shorter period may belong to two longer periods simultaneously, reflecting the complexity of the time series. There is a clear similarity between components within the same level and those in adjacent levels where inclusion relationships exist. Thus inspired by Pyraformer [10], we propose the **Peri**odic Pyra**mid** Trans**former** (**Peri-midFormer**), which computes self-attention among periodic components to capture complex temporal variations in time series. Furthermore, we consider each branch in the Periodic Pyramid as a Periodic Feature Flow, and aggregating features from multiple flows to provide rich periodic information for downstream tasks. In experiments, Peri-midFormer achieves state-of-the-art performance in various analytic tasks, including forecasting, imputation, anomaly detection, and classification.

1. Based on the inclusion relationships of multiple periods in time series, this paper proposes a top-down constructed Periodic Pyramid structure, which expands 1D time series variations into 2D, explicitly representing the implicit multi-period relationships within the time series.

2. We propose Peri-midFormer, which uses the Periodic Pyramid Attention Mechanism to automatically capture dependencies between different and same-level periodic components, extracting diverse temporal variations in time series. Additionally, to further harness the potential of Peri-midFormer, we introduce Periodic Feature Flows to provide rich periodic information for downstream tasks.

3. We conduct extensive experiments on five mainstream time series analysis tasks, and Peri-midFormer achieves state-of-the-art across all of them, demonstrating its superior capability in time series analysis.

The remainder of this paper is structured as follows. Section 2 briefly summarizes the related work. Section 3 details the proposed model structure. Section 4 extensively evaluates our method's performance across five main time series analysis tasks. Section 5 presents ablations analysis, Section 6 presents complexity analysis, and Section 7 discusses our results and future directions.

## 2    Related Work

Temporal variation modeling, a crucial aspect of time series analysis, has been extensively investigated. In recent years, numerous deep models have emerged for this purpose, including MLP [14; 15], TCN [8], and RNN [6; 7]-based architectures. Furthermore, Transformers have shown remarkable performance in time series forecasting [16; 12; 17; 18]. They utilize attention mechanisms to uncover temporal dependencies among time points. For instance, Wu et al. [12] introduce Autoformer with an Auto-Correlation mechanism, adept at capturing series-wise temporal dependencies derived from learned periods. Moreover, to address complex temporal patterns, they adopted a deep decomposition architecture to extract seasonal and trend parts from input series. Subsequently, FEDformer [17] enhances seasonal-trend decomposition through a mixture-of-expert design and introduces sparse attention within the frequency domain. Pyraformer [10] constructs a down-top pyramid structure through multiple convolution operations on time series to address the issue of long information propagation paths in Transformers, significantly reducing both time and space complexity. PatchTST [19] partitions individual data points into patches and uses them as tokens for the Transformer, thereby enhancing its understanding of local information in time series. Additionally, PatchTST innovatively processes each channel separately, making it particularly suitable for forecasting tasks.

Additionally, there are some recent innovative works. Timesnet [13] unravels intricate temporal patterns by exploring the multi-periodicity of time series and captures temporal 2D-variations using computer vision CNN backbones. GPT4TS [20] ingeniously utilizes the large language model GPT2 as a pretrained model, fine-tuning some of its structures with time series, achieving state-of-the-art results. FITS [21] proposes a time series analysis model based on frequency domain operations, requiring very low parameter count and memory consumption. And recent works have considered multi-scale information in time series. PDF [22] captures both short-term and long-term variations by transforming 1D time series into 2D tensors using a multi-periodic decoupling block. It achieves superior forecasting performance by modeling these decoupled variations and integrating them for accurate predictions. SCNN [23] decomposes multivariate time series into long-term, seasonal, short-term, co-evolving, and residual components, enhancing interpretability, adaptability to distribution shifts, and scalability by modeling each component separately. TimeMixer [24] uses a novel multiscale mixing approach, decomposing time series into fine and coarse scales to capture both detailed and macroscopic variations. It employs Past-Decomposable-Mixing to extract historical information and Future-Multipredictor-Mixing to leverage multiscale forecasting capabilities, achieving great performance in forecasting task.

## 3    Methodology

### 3.1    Model Structure

The overall flowchart of the proposed approach is shown in Figure 2, it begins with time embedding of the original time series at the top. Then, we use the FFT to decompose it into multiple periodic components of varying lengths across different levels, with lines indicating the inclusion relationships between them. Moving down, padding and projection are then applied to ensure uniform dimensions, forming the Periodic Pyramid. Each component is treated as an independent token and receives positional embedding. Next, the Periodic Pyramid is fed into Peri-midFormer, which consists of multiple layers for computing Periodic Pyramid Attention. Finally, depending on the task, two strategies are employed: for classification, components are directly concatenated and projected into the category space; for other reconstruction tasks (since forecasting, imputation, and anomaly detection all necessitate the model to reconstruct the channel dimensions or input lengths, we collectively refer to such tasks as reconstruction tasks), features from different pyramid branches are integrated through Periodic Feature Flows Aggregation to generate the final output. Please note that we referred to [11] for de-normalization and [12] for time series decomposition to maximize the effectiveness of our method, but we omitted these details from the figure to maintain simplicity. See Appendix A for a complete flowchart. Further details are provided below.

### 3.2    Periodic Pyramid Construction

Multiple periods in the time series exhibit clear inclusion relationships, however, the 1D structure limits the representation of variations between them. Hence, it's crucial to separate periodic com-

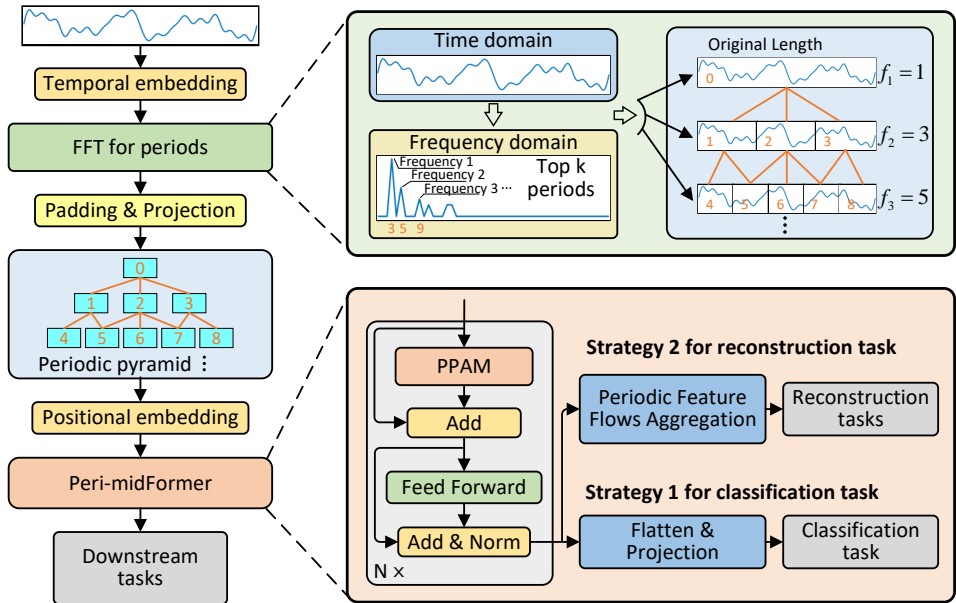

Figure 2: Model architecture. PPAM denotes Periodic Pyramid Attention Mechanism.

ponents with inclusion relationships to explicitly represent implicit periodic relationships. Firstly, as Peri-midFormer is designed to focus on periodic components, we first normalize the original time series $\mathbf{X} \in \mathbb{R}^{L \times C}$ that with length $L$ and $C$ channels, then decompose it to obtain the seasonal part $\mathbf{X}_s \in \mathbb{R}^{L \times C}$, thus removing the interference of the trend part. For a detailed description of normalization and decomposition, please refer to the Appendix A. Then we partition $\mathbf{X}_s$ into periodic components, following the approach used in Timesnet [13]. It's important to note that, inspired by PatchTST [19], we retain the channel dimension $C$, as it is advantageous for Peri-midFormer in capturing periodic features within each channel (note that we adopt a channel independent strategy and the Figure 2 shows the processing of only one of the channels). The periodic components are extracted in the frequency domain, accomplished specifically through FFT:

$$\mathbf{A} = Avg\left(Amp\left(FFT\left(\mathbf{X}_s\right)\right)\right), \{f_1, \cdots, f_k\} = \underset{f_* \in \left\{1, \cdots, \left\lceil \frac{L}{2} \right\rceil\right\}}{\arg \operatorname{Topk}} (\mathbf{A}), p_i = \left\lceil \frac{L}{f_i} \right\rceil, i \in \{1, \cdots, k\}, \quad (1)$$

where $FFT(\cdot)$ and $Amp(\cdot)$ denote Fourier Transform and amplitude calculation, respectively. $\mathbf{A} \in \mathbb{R}^L$ represents the amplitude of each frequency, averaged across $C$ channels using $Avg(\cdot)$. Note that the $j$-th value $\mathbf{A}j$ denotes the intensity of the $j$-th frequency periodic basis function, associated with period length $\left\lceil \frac{L}{j} \right\rceil$. To handle frequency domain sparsity and minimize noise from irrelevant high frequencies [17], the top-$k$ amplitude values $\{\mathbf{A}_{f_1}, \cdots, \mathbf{A}_{f_k}\}$ corresponding to the most significant frequencies $\{f_1, \cdots, f_k\}$ are selected, where $k$ is a hyper-parameter, beginning from 2 to ensure the fundamental pyramid structure. Additionally, to ensure the top level of the pyramid corresponds to the original time series, we define $f_1 = 1$, with other frequencies arranged in ascending order. These selected frequencies correspond to $k$ period lengths $\{p_1, \cdots, p_k\}$, arranged in descending order. Due to the frequency domain's conjugacy, only frequencies within $\left\{1, \cdots, \left\lceil \frac{L}{2} \right\rceil\right\}$ are considered. Based on the selected frequencies $\{f_1, \cdots, f_k\}$ and their associated period lengths $\{p_1, \cdots, p_k\}$, we partition the original time series into periodic components for each pyramid level, denoted as $\mathbf{C}_\ell$:

$$\mathbf{C}_\ell = \{\mathbf{C}_\ell^1, \mathbf{C}_\ell^2, \cdots, \mathbf{C}_\ell^n\}, \ell \in \{1, \cdots, k\}, n \in \{1, \cdots, f_k\}, \quad (2)$$

where $\mathbf{C}_\ell^n$ denotes the $n$-th periodic component in the $\ell$-th pyramid level. Here, $\ell$ is the pyramid level index, starting from the top and increasing, with a maximum value of $k$, indicating the number of levels determined by the selected periods. Similarly, $n$ represents the component index within a level, increasing from left to right, with a maximum value of $f_k$, indicating the number of components per level is determined by the frequency corresponding to that period in the original series. The Periodic Pyramid can thus be represented as:

$$\mathbf{P} = Stack\left(\mathbf{C}_\ell\right), \ell \in \{1, \cdots, k\}, \quad (3)$$

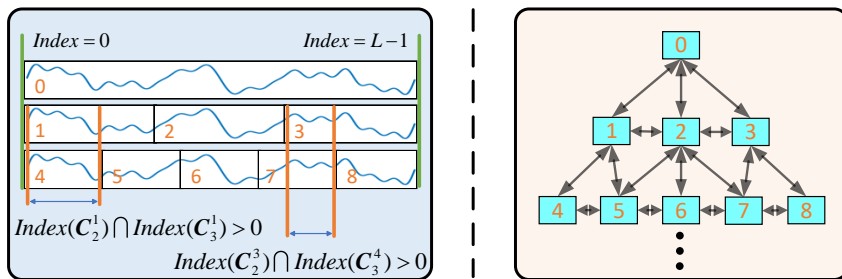

Figure 3: Inclusion relationships of periodic components (left) and Periodic Pyramid Attention Mechanism (right).

where $Stack(\cdot)$ denotes a stacking operation. The constructed Periodic Pyramid, depicted in the upper right of Figure 2, displays evident inclusion relationship among different levels, shown by connections between levels. Let $R$ denote the relationship between pairs of periodic components from the upper and lower levels, determined by the presence or absence of overlap as follows:

$$R_{\ell-1,\ell}^{n_{\ell-1},n_\ell} = \begin{cases} 1, & Index(\mathbf{C}_{\ell-1}^{n_{\ell-1}}) \bigcap Index(\mathbf{C}_\ell^{n_\ell}) > 0 \\ 0, & else \end{cases}, \ell \in \{2, \cdots, k\}, n_{\ell-1}, n_\ell \in \{1, \cdots, f_k\}, \quad (4)$$

when $R = 1$, it signifies an inclusion relationship, while $R = 0$ indicates no overlap. This is illustrated in the left half of Figure 3. $n_\ell$ denotes the index of the $n$-th component in the $\ell$-th level. $Index(\cdot)$ denotes the positional index of each data point within the periodic component at that level. Indices for points in the first level are contained in $\{0, \ldots, L-1\}$. For subsequent levels, most indices match those of the first level. However, due to varying component lengths, there may be slight differences in indices for the last portion. Nonetheless, this doesn't impact relationship determination between components across levels. In practice, the relationship between the components is realized by masking the corresponding elements in the attention matrix.

Thanks to the inclusion relationships between periodic components across different levels in the Periodic Pyramid, complex periodic relationships inherent in 1D time series are explicitly represented. Next, due to the varying lengths of the components, it's necessary to map $\mathbf{C}^\ell$ to the same scale for subsequent Periodic Pyramid Attention Mechanism, with the equation provided as follows:

$$\mathbf{P}' = Projection\left(Padding\left(\mathbf{C}_\ell^n\right)\right), \ell \in \{1, \cdots, k\}, n \in \{1, \cdots, f_k\}, \quad (5)$$

where $Padding(\cdot)$ denotes zero-padding the periodic components across the time dimension to match the length of the original data, while $Projection(\cdot)$ represents a single linear mapping layer.

### 3.3 Periodic Pyramid Transformer (Peri-midFormer)

Once we have the Periodic Pyramid, it can be inputted into the Peri-midFormer, as depicted in Figure 2. The Peri-midFormer introduces a specialized attention mechanism tailored for the Periodic Pyramid, called **P**eriodic **P**yramid **A**ttention **M**echanism (**PPAM**), shown in the right half of Figure 3. Here, original connections are replaced with bidirectional arrows, and also added within the same level. These bidirectional arrows signify attention between periodic components. In PPAM, inter-level attention focuses on period dependencies across levels, while intra-level attention focuses on dependencies within the same level. Note that attention occurs among all components within the same level, not just between adjacent ones. However, for clarity, not all attention connections within the same level are depicted.

In Periodic Pyramid, a periodic component $\mathbf{C}_\ell^n$ generally has three types of interconnected relationships (denoted as $\mathbb{I}$) with other components: the parent node in the level above (denoted as $\mathbb{P}$), all nodes within the same level including itself (denoted as $\mathbb{A}$), and the child nodes in its next level (denoted as $\mathbb{C}$). Therefore, the relationships of $\mathbf{C}_\ell^n$ can be expressed by the following equation:

$$\begin{cases} \mathbb{I}_\ell^{(n)} = \mathbb{P}_\ell^{(n)} \bigcup \mathbb{A}_\ell^{(n)} \bigcup \mathbb{C}_\ell^{(n)} \\ \mathbb{P}_\ell^{(n)} = \left\{ \mathbf{C}_{\ell-1}^j : j = \{n_{\ell-1} : R_{\ell-1,\ell}^{n_{\ell-1},n_\ell} = 1\} \right\}, & \text{if } \ell \geq 2 \text{ else } \varnothing \\ \mathbb{A}_\ell^{(n)} = \left\{ \mathbf{C}_\ell^j : 1 \leq j \leq f_k \right\} \\ \mathbb{C}_\ell^{(n)} = \left\{ \mathbf{C}_{\ell+1}^j : j = \{n_{\ell+1} : R_{\ell,\ell+1}^{n_\ell,n_{\ell+1}} = 1\} \right\}, & \text{if } \ell \leq k-1 \text{ else } \varnothing \end{cases}. \quad (6)$$

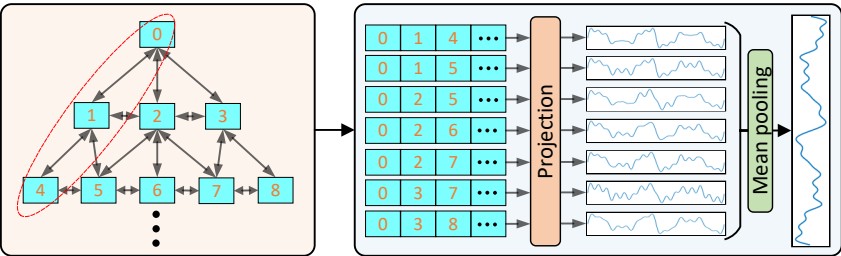

Figure 4: Periodic Feature Flows Aggregation.

The equation shows that a component at the topmost level lacks a parent node, while one at the bottommost level lacks a child node. Based on the interconnected relationships $\mathbb{I}$, the attention of the component $\mathbf{C}_\ell^n$ can be expressed as:

$$\mathbf{a}_i = \sum_{m \in \mathbb{I}_\ell^{(n)}} \frac{\exp\left(\mathbf{q}_i \mathbf{k}_m^\top / \sqrt{d_K}\right) \mathbf{v}_m}{\sum\limits_{m \in \mathbb{I}_\ell^{(n)}} \exp\left(\mathbf{q}_i \mathbf{k}_m^\top / \sqrt{d_K}\right)}, \tag{7}$$

where $\mathbf{q}$, $\mathbf{k}$, and $\mathbf{v}$ denote query, key, and value vectors, respectively, as in the classical self-attention mechanism. $m$ used for selecting components that have interconnected relationships with $\mathbf{C}_\ell^n$. $\mathbf{k}_m^\top$ represents the transpose of row $m$ in $K$. $d_K$ refers to the dimension of key vectors, ensuring stable attention scores through scaling.

We apply this attention mechanism to each component across all levels of the Periodic Pyramid, enabling the automatic detection of dependencies among all components in the Periodic Pyramid and capturing the intricate temporal variations in the time series. For a detailed theoretical proof of the PPAM see the Appendix F.

### 3.4 Periodic Feature Flows Aggregation

Here we explain the Periodic Feature Flows Aggregation used for reconstruction tasks. The output of the Peri-midFormer retains the original pyramid structure. To leverage the diverse periodic components across different levels, we treat a single branch from the top to the bottom of the pyramid as a periodic feature flow, highlighted by the red line in Figure 4. Since a periodic feature flow passes through periodic components at different levels, it contains periodic features of different scales from the time series. Additionally, due to variations among periodic components within each level, each feature flow carries distinct information. Therefore, we aggregate multiple feature flows through Periodic Feature Flow Aggregation. This involves linearly mapping each feature flow to match the length of the target time series and then averaging it across multiple feature flows to obtain the aggregated result $\mathbf{Y}_s$, as expressed in the following equation:

$$\mathbf{Y}_s = MeanPolling\left(Projection\left(\left\{\hat{\mathbf{C}}_1^{n_1}, \hat{\mathbf{C}}_2^{n_2}, \cdots, \hat{\mathbf{C}}_\ell^{n_k}\right\}\right)\right), \ell \in \{2, \cdots, k\}, n_k \in \{1, \cdots, f_k\}, \tag{8}$$

where $\hat{\mathbf{C}}_\ell^{n_k}$ represents a specific periodic component in the Peri-midFormer's output, and $\{\hat{\mathbf{C}}_1^{n_1}, \hat{\mathbf{C}}_2^{n_2}, \cdots, \hat{\mathbf{C}}_\ell^{n_k}\}$ forms a feature flow, as shown in Figure 4. $Projection(\cdot)$ maps each feature flow to match the target output length. $Meanpooling(\cdot)$ averages the feature flows. $\mathbf{Y}_s$ indicates that this is the output from the seasonal part. Since we retained the channel dimension of the original time series, the result obtained after aggregating the periodic feature flows here becomes the shape of the final output. Finally, adding the trend part and de-normalization to obtain the ultimate output.

## 4 Experiments

We extensively test Peri-midFormer on five mainstream analysis tasks: short- and long-term fore-casting, imputation, classification, and anomaly detection. We adopted the same benchmarks as Timesnet [13], see Appendix C for details. Due to space limits, we provide only a summary of the results here, more details about the datasets, experiment implementation, model configuration, and full results can be found in Appendix.

**Baselines** The baselines include CNN-based models: TimesNet [13]; MLP-based models: LightTS [15], DLinear [14] and FITS [21]; Transformer-based models: GPT4TS [20], Time-LLM [25], iTransformer [26], TSLANet [27], Reformer [28], Pyraformer [10], Informer [16], Autoformer [12], FEDformer [17], Non-stationary Transformer [11], ETSformer [29], PatchTST [19]. Besides, N-HiTS [30] and N-BEATS [31] are used for short-term forecasting. Anomaly Transformer [32] is used for anomaly detection. Rocket [33], LSTNet [34], TCN [8] and Flowformer [35] are used for classification.

## 4.1 Main Results

Figure 5 displays the comprehensive comparison results between Peri-midFormer and other methods, it consistently excels across all five tasks.

## 4.2 Long-term Forecasting

**Setups** Referring to [13], we adopt eight real-world benchmark datasets for long-term forecasting, including Weather [36], Traffic [37], Electricity [38], Exchange [34], and four ETT [16] datasets (ETTh1, ETTh2, ETTm1, ETTm2). Forecast lengths are set to 96, 192, 336, and 720. For the fairness of the comparison, we set the look-back window for all the methods to 512 (64 on Exchange), the results for other look-back windows can be found in the Appendix H.3

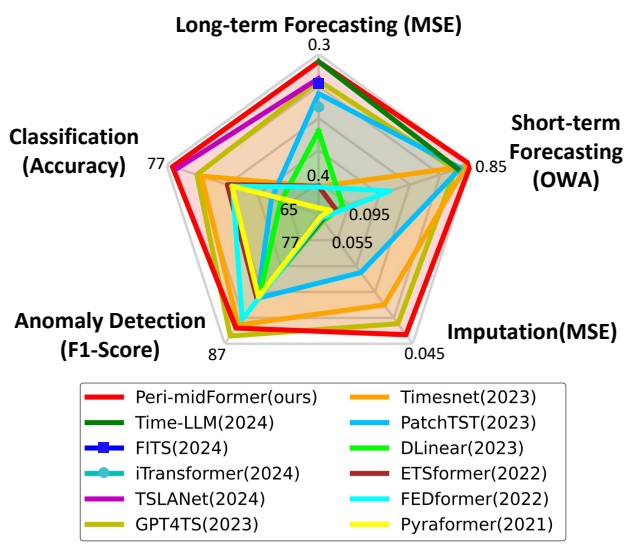

Figure 5: Model performance comparison.

Table 1: Long-term forecasting task. The results are averaged from four different series length {96, 192, 336, 720}. See Table 13 and 14 for full results. **Red**: best, Blue: second best.

| Models | Peri-mid Former | | GPT4TS [20] | | TSLANet [27] | | Time-LLM [25] | | FITS [21] | | DLinear [14] | | PatchTST [19] | | TimesNet [13] | | Pyraformer [10] | |
|---|---|---|---|---|---|---|---|---|---|---|---|---|---|---|---|---|---|---|
| Metric | MSE | MAE | MSE | MAE | MSE | MAE | MSE | MAE | MSE | MAE | MSE | MAE | MSE | MAE | MSE | MAE | MSE | MAE |
| Weather | 0.233 | 0.271 | 0.237 | 0.270 | 0.276 | 0.291 | **0.225** | **0.257** | 0.244 | 0.281 | 0.241 | 0.294 | 0.226 | 0.266 | 0.249 | 0.286 | 0.281 | 0.349 |
| ETTh1 | 0.409 | 0.430 | 0.427 | 0.426 | 0.422 | 0.440 | 0.408 | 0.423 | **0.407** | **0.422** | 0.418 | 0.438 | 0.430 | 0.444 | 0.492 | 0.490 | 0.913 | 0.748 |
| ETTh2 | **0.317** | **0.377** | 0.354 | 0.394 | 0.328 | 0.385 | 0.334 | 0.383 | 0.333 | 0.382 | 0.504 | 0.482 | 0.388 | 0.414 | 0.408 | 0.440 | 3.740 | 1.554 |
| ETTm1 | 0.354 | 0.385 | 0.352 | 0.383 | 0.348 | 0.383 | **0.329** | **0.372** | 0.358 | 0.378 | 0.356 | 0.379 | 0.363 | 0.391 | 0.398 | 0.418 | 0.724 | 0.609 |
| ETTm2 | 0.258 | 0.320 | 0.266 | 0.326 | 0.263 | 0.325 | **0.251** | **0.313** | 0.254 | **0.313** | 0.275 | 0.342 | 0.273 | 0.329 | 0.287 | 0.343 | 1.356 | 0.797 |
| Electricity | **0.152** | 0.249 | 0.167 | 0.263 | 0.159 | **0.224** | 0.158 | 0.252 | 0.168 | 0.263 | 0.167 | 0.268 | 0.162 | 0.257 | 0.217 | 0.314 | 0.299 | 0.391 |
| Traffic | 0.392 | 0.270 | 0.414 | 0.294 | 0.397 | 0.272 | **0.388** | **0.264** | 0.420 | 0.287 | 0.433 | 0.305 | 0.392 | 0.270 | 0.622 | 0.332 | 0.705 | 0.401 |
| Exchange | **0.346** | **0.393** | 0.373 | 0.410 | 0.365 | 0.410 | 0.371 | 0.409 | 0.393 | 0.429 | 0.495 | 0.493 | 0.418 | 0.433 | 0.701 | 0.593 | 1.157 | 0.844 |
| Average | **0.308** | 0.337 | 0.324 | 0.346 | 0.320 | 0.341 | **0.308** | **0.334** | 0.322 | 0.344 | 0.361 | 0.375 | 0.331 | 0.350 | 0.422 | 0.402 | 1.147 | 0.711 |

**Results** From Table 1, it is evident that Peri-midFormer performs exceptionally well, even completely outperforms GPT4TS and closely approaching the state-of-the-art method Time-LLM. While Time-LLM demonstrates remarkable capabilities in long-term forecasting, our Peri-midFormer shows clear advantages on the ETTh2, Electricity, and Exchange datasets. Although Time-LLM achieves the best results, it relies on a very large model, leading to significant computational overhead that is unavoidable. The same issue exists for GPT4TS. In contrast, our Peri-midFormer achieves performance close to that of Time-LLM without requiring excessive computational resources, making it more suitable for practical applications. Further analysis of model complexity is provided in Section 6. In addition, our Peri-midFormer exhibits better performance with longer look-back window, as further detailed in the Appendix E.6.

Table 2: Short-term forecasting task on M4. The prediction lengths are in $\{6, 48\}$ and results are weighted averaged from several datasets under different sample intervals. ($*$ means former, Station means the Non-stationary Transformer.) See Table 12 for full results. **Red**: best, Blue: second best.

| Models | Peri-mid Former | Time-LLM [25] | GPT4TS [20] | TimesNet [13] | PatchTST [19] | N-HiTS [40] | N-BEATS [31] | ETS* [29] | LightTS [41] | DLinear [14] | FED* [17] | Station [11] | Auto* [12] | Pyra* [10] | In* [16] | Re* [28] |
|---|---|---|---|---|---|---|---|---|---|---|---|---|---|---|---|---|
| SMAPE | 11.833 | 11.983 | 11.991 | **11.829** | 12.059 | 11.927 | 11.851 | 14.718 | 13.525 | 13.639 | 12.840 | 12.780 | 12.909 | 16.987 | 14.086 | 18.200 |
| MASE | **1.584** | 1.595 | 1.600 | 1.585 | 1.623 | 1.613 | 1.599 | 2.408 | 2.111 | 2.095 | 1.701 | 1.756 | 1.771 | 3.265 | 2.718 | 4.223 |
| OWA | **0.850** | 0.859 | 0.861 | 0.851 | 0.869 | 0.861 | 0.855 | 1.172 | 1.051 | 1.051 | 0.918 | 0.930 | 0.939 | 1.480 | 1.230 | 1.775 |

## 4.3 Short-term Forecasting

**Setups** For short-term analysis, we adopt the M4 [39], which contains the yearly, quarterly and monthly collected univariate marketing data. We measure forecast performance using the symmetric mean absolute percentage error (SMAPE), mean absolute scaled error (MASE), and overall weighted average (OWA), which are calculated as detailed in the Appendix D.1.

**Results** Table 2 shows that Peri-midFormer outperforms Time-LLM, GPT4TS, TimesNet, and N-BEATS, highlighting its exceptional performance in short-term forecasting. In the M4 dataset, some data lacks clear periodicity, such as the Yearly data, which mainly exhibits a strong trend. A similar situation is observed in the Exchange dataset for long-term forecasting task. Peri-midFormer performs well on these datasets due to its time series decomposition strategy. For a detailed analysis, please refer to the Appendix E.7.

## 4.4 Time Series Classification

**Setups** We assessed Peri-midFormer's capacity for high-level representation learning via classification task. Mimicking settings akin to TimesNet [13], we tested it on 10 multivariate UEA classification datasets from [42], covering tasks like gesture recognition, action recognition, audio recognition, medical diagnoses, and other real-world applications.

**Results** As shown in Figure 6, Peri-midFormer achieves an average accuracy of 76.6%, surpassing all baselines including TSLANet (76.0%), GPT4TS (74.0%), Times-Net (73.6%), and all other Transformer-based methods. This suggests that Peri-midFormer has excellent time series representation capabilities. See Appendix H.1 for full results.

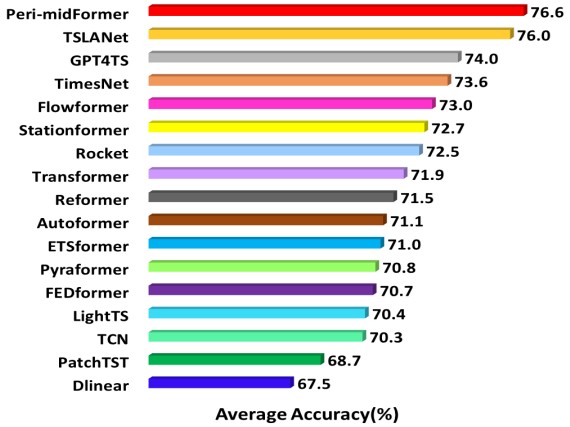

Figure 6: Model comparison in classification. The results are averaged from 10 subsets of UEA.

## 4.5 Imputation

**Setups** To validate Peri-midFormer's imputation capabilities, we conduct experiment on six real-world datasets, including four ETT datasets [16] (ETTh1, ETTh2, ETTm1, ETTm2), Electricity [38], and Weather [36]. We evaluate different random mask ratios (12.5%, 25%, 37.5%, 50%) for varying levels of missing data. Notably, due to the large number of missing values, the time series do not reflect their original periodicity. Therefore, before imputation, we simply interpolate the original missing data through a linear interpolation strategy in order to use Peri-midFormer efficiently, which we call pre-interpolation. For a description of pre-interpolation and its impact on other methods, please refer to the Appendix E.5.

**Results** Table 3 demonstrates Peri-midFormer's outstanding performance on specific datasets (Electricity and Weather), surpassing other methods significantly and securing the highest average scores. However, its performance on other datasets was ordinary, possibly due to the lack of obvious periodic characteristics in them.

Table 3: Imputation task. We randomly mask $\{12.5\%, 25\%, 37.5\%, 50\%\}$ time points of length-96 time series. The results are averaged from 4 different mask ratios. (∗ means former, Station means the Non-stationary Transformer.) See Table 15 for full results. **Red**: best, Blue: second best.

| Models | Peri-mid Former | | GPT4TS [20] | | TimesNet [13] | | PatchTST [19] | | ETS∗ [29] | | LightTS [15] | | DLinear [14] | | FED∗ [17] | | Station [11] | | Auto∗ [12] | | Pyra∗ [10] | |
|---|---|---|---|---|---|---|---|---|---|---|---|---|---|---|---|---|---|---|---|---|---|---|
| Metric | MSE | MAE | MSE | MAE | MSE | MAE | MSE | MAE | MSE | MAE | MSE | MAE | MSE | MAE | MSE | MAE | MSE | MAE | MSE | MAE | MSE | MAE |
| ETTm1 | 0.034 | 0.116 | 0.028 | **0.105** | **0.027** | 0.107 | 0.047 | 0.140 | 0.120 | 0.253 | 0.104 | 0.218 | 0.093 | 0.206 | 0.062 | 0.177 | 0.036 | 0.126 | 0.051 | 0.150 | 0.717 | 0.570 |
| ETTm2 | 0.025 | 0.087 | **0.021** | **0.084** | 0.022 | 0.088 | 0.029 | 0.102 | 0.208 | 0.327 | 0.046 | 0.151 | 0.096 | 0.208 | 0.101 | 0.215 | 0.026 | 0.099 | 0.029 | 0.105 | 0.465 | 0.508 |
| ETTh1 | 0.083 | 0.191 | **0.069** | **0.173** | 0.078 | 0.187 | 0.115 | 0.224 | 0.202 | 0.329 | 0.284 | 0.373 | 0.201 | 0.306 | 0.117 | 0.246 | 0.094 | 0.201 | 0.103 | 0.214 | 0.842 | 0.682 |
| ETTh2 | 0.054 | 0.146 | **0.048** | **0.141** | 0.049 | 0.146 | 0.065 | 0.163 | 0.367 | 0.436 | 0.119 | 0.250 | 0.142 | 0.259 | 0.163 | 0.279 | 0.053 | 0.152 | 0.055 | 0.156 | 1.079 | 0.792 |
| Electricity | **0.060** | **0.160** | 0.090 | 0.207 | 0.092 | 0.210 | 0.072 | 0.183 | 0.214 | 0.339 | 0.131 | 0.262 | 0.132 | 0.260 | 0.130 | 0.259 | 0.100 | 0.218 | 0.101 | 0.225 | 0.297 | 0.382 |
| Weather | **0.028** | **0.039** | 0.031 | 0.056 | 0.030 | 0.054 | 0.060 | 0.144 | 0.076 | 0.171 | 0.055 | 0.117 | 0.052 | 0.110 | 0.099 | 0.203 | 0.032 | 0.059 | 0.031 | 0.057 | 0.152 | 0.235 |
| Average | **0.047** | **0.123** | 0.048 | 0.128 | 0.050 | 0.132 | 0.053 | 0.159 | 0.164 | 0.309 | 0.123 | 0.229 | 0.119 | 0.225 | 0.112 | 0.230 | 0.057 | 0.143 | 0.062 | 0.151 | 0.592 | 0.528 |

## 4.6 Time Series Anomaly Detection

**Setups** For anomaly detection, we assess models on five standard datasets: SMD [43], MSL [44], SMAP [44], SWaT [45] and PSM [46]. To ensure fairness, we exclusively use classical reconstruction error for all baseline models, aligning with the approach in TimesNet [13]. Specifically, normal data is used for training, and a simple reconstruction loss is employed to help the model learn the distribution of normal data. In the testing phase, parts of the reconstructed output that exceed a certain threshold are considered anomalies. We use a point adjustment technique combined with a manually set threshold for this purpose.

Table 4: Anomaly detection task. We calculate the F1-score (as %) for each dataset. (∗ means former, Station means the Non-stationary Transformer.) A higher value of F1-score indicates a better performance. See Table 16 for full results. **Red**: best, Blue: second best.

| Models | Peri-mid Former | GPT4TS [20] | TimesNet [13] | PatchTST [19] | ETS∗ [29] | FED∗ [17] | LightTS [15] | DLinear [14] | Station [11] | Auto∗ [12] | Pyra∗ [10] | Ano∗ [32] | In∗ [16] | Re∗ [28] | LogTrans∗ [47] | Trans∗ [48] |
|---|---|---|---|---|---|---|---|---|---|---|---|---|---|---|---|---|
| SMD | 85.95 | **86.89** | 84.61 | 84.62 | 83.13 | 85.08 | 82.53 | 77.1 | 84.62 | 85.11 | 83.04 | 85.49 | 81.65 | 75.32 | 76.21 | 79.56 |
| MSL | 81.83 | 82.45 | 81.84 | 78.7 | **85.03** | 78.57 | 78.95 | 84.88 | 77.5 | 79.05 | 84.86 | 83.31 | 84.06 | 84.40 | 79.57 | 78.68 |
| SMAP | 68.62 | **72.88** | 69.39 | 68.82 | 69.50 | 70.76 | 69.21 | 69.26 | 71.09 | 71.12 | 71.09 | 71.18 | 69.92 | 70.40 | 69.97 | 69.70 |
| SWaT | 93.40 | **94.23** | 93.02 | 85.72 | 84.91 | 93.19 | 93.33 | 87.52 | 79.88 | 92.74 | 91.78 | 83.10 | 81.43 | 82.80 | 80.52 | 80.37 |
| PSM | 97.19 | 97.13 | **97.34** | 96.08 | 91.76 | 97.23 | 97.15 | 93.55 | 97.29 | 93.29 | 82.08 | 79.40 | 77.10 | 73.61 | 76.74 | 76.07 |
| Avg F1 | 85.40 | **86.72** | 85.24 | 82.79 | 82.87 | 84.97 | 84.23 | 82.46 | 82.08 | 84.26 | 82.57 | 80.50 | 78.83 | 77.31 | 76.60 | 76.88 |

**Results** The results in Table 4 illustrate that Peri-midFormer's anomaly detection capability is second only to GPT4TS, but a large gap does exist. This is due to the binary event data nature in the anomaly detection dataset [21], which makes it difficult for Peri-midFormer to capture useful periodic characteristics, leading to general performance.

## 5 Ablations

Table 5: Ablation experiments on long-term forecasting task to verify the effect of each component. **Red**: best, Blue: second best.

| Datasets / Variant | ETTh1 | | ETTh2 | | Electricity | | Weather | | Traffic | |
|---|---|---|---|---|---|---|---|---|---|---|
| | MSE | MAE | MSE | MAE | MSE | MAE | MSE | MAE | MSE | MAE |
| Pyraformer | 0.913 | 0.748 | 0.826 | 0.703 | 0.299 | 0.391 | 0.281 | 0.349 | 0.705 | 0.401 |
| w/o period components | 0.734 | 0.781 | 0.407 | 0.442 | 0.212 | 0.345 | 0.261 | 0.333 | 0.499 | 0.391 |
| w/o PPAM | 0.581 | 0.599 | 0.344 | 0.393 | 0.164 | 0.259 | 0.254 | 0.291 | 0.415 | 0.364 |
| w/o Feature Flows Aggregation | 0.433 | 0.447 | 0.341 | 0.391 | 0.155 | 0.250 | 0.244 | 0.280 | 0.397 | 0.277 |
| **Peri-midFormer** | **0.409** | **0.430** | **0.317** | **0.377** | **0.152** | **0.249** | **0.233** | **0.271** | **0.391** | **0.269** |

**Setups** We performed ablation experiments on key modules of Peri-midFormer for the long-term forecasting task, with results presented in Table 5. The table outlines the progression of module additions, from top to bottom. "Pyraformer" refers to the Pyraformer [10], building the pyramid down-top with convolutions and employing a simple two-fold relationship for attention distribution. "w/o periodic components" constructs a pyramid top-down by dividing the time series into patches without

Classification on the UEA Heartbeat

Long-term forecasting of length 720 on ETTh2

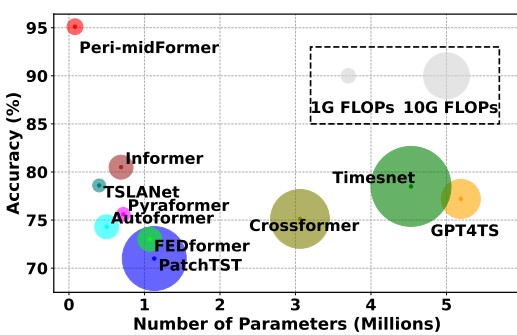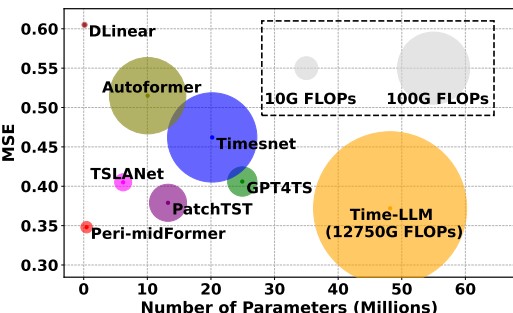

Figure 7: Number of training parameters and FLOPs for Peri-midFormer versus baseline in terms of classification accuracy for the UEA Heartbeat dataset (left) and long-term forecasting MSE for the ETTh2 dataset (right). In the left graph, the closer to the top left, the better, while in the right graph, the closer to the bottom left, the better. Note that in the long-term forecasting, we did not fully depict the corresponding sizes due to the oversized FLOPs of Time-LLM, but instead illustrated it with text.

considering periodic components. "w/o PPAM" divides the time series into periodic components but without Period Pyramid Attention Mechanism, using periodic full attention instead, wherein attention is computed among all periodic components. "w/o Feature Flows Aggregation" employs PPAM but without Periodic Feature Flows Aggregation. "Peri-midFormer" indicates our final approach.

**Results** Table 5 illustrates the incremental performance enhancement achieved with the integration of each additional module, validating the effectiveness of Peri-midFormer. Notably, good results are achieved even without PPAM. This can be attributed to the model's ability to extract periodic characteristics inherent in the original time series data by delineating the periodic components. However, without highlighting inclusion relationships through PPAM, periodic full attention's ability to capture temporal changes is limited, emphasizing the significance of PPAM.

## 6 Complexity Analysis

We conducted experiments on the model complexity of Peri-midFormer using the Heartbeat dataset for the classification task and the ETTh2 dataset for the long-term forecasting task. We considered the number of training parameters, FLOPs, and accuracy (or MSE). The results are depicted in Figure 7. In the classification task, Peri-midFormer not only achieves a significant advantage in accuracy but also requires relatively fewer training parameters and FLOPs, much less than many methods such as TimesNet, GPT4TS, Crossformer, and PatchTST. In the long-term forecasting task, Peri-midFormer achieves the lowest MSE without requiring the enormous FLOPs that Time-LLM does. This shows that although Time-LLM has strong long-term forecasting capabilities on most datasets, its computational demands are unacceptable. See Appendix E.4 for more analysis.

Further Model Analysis is provided in the Appendix E.

## 7 Conclusions

In this paper, we introduced a method for general time series analysis called Peri-midFormer. It leverages the multi-periodicity of time series and the inclusion relationships between different periods. By segmenting the original time series into different levels of periodic components, Peri-midFormer constructs a Periodic Pyramid along with its corresponding attention mechanism. Through extensive experiments covering forecasting, classification, imputation, and anomaly detection tasks, we validated the capabilities of Peri-midFormer in time series analysis, achieving outstanding results across all tasks. However, Peri-midFormer exhibits limitations, particularly in scenarios where the periodic characteristics are less apparent. We aim to address this limitation in future research to broaden its applicability.

## Acknowledgement

This work was supported by the National Natural Science Foundation of China under Grant Nos. 62276205, U22B2018, and Graduate Student Innovation Fund under Grant Nos. YJSJ24012.

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

## A    Method Details

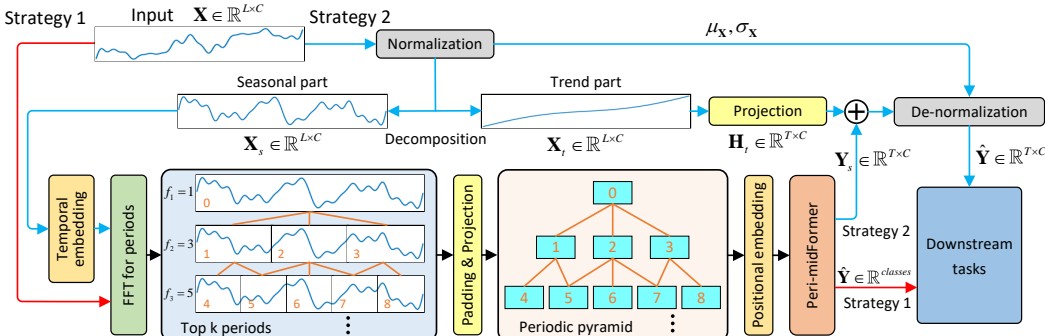

Figure 8: Full flowchart.

Here, we provide further elaboration on the details of Peri-midFormer. Its full flowchart is depicted in Figure 8, depicts two strategies for input. The strategy indicated by the red line is for classification task, where de-normalization and time series decomposition are not used. This is because, in classification task, there is no need to reconstruct the original data; therefore, the trend part does not need to be extracted and added back. Additionally, it is important to note that the trend part is a significant discriminative feature for classification data, so it cannot be separated from the original data before feature extraction. In the strategy indicated by the blue line, employed for reconstruction tasks, Peri-midFormer needs to focus on the periodicity in the time series. Therefore, we utilize de-normalization and time series decomposition to eliminate other influencing factors. To achieve this, we first refer to [11] to address instability factors. We normalize the input $\mathbf{X} = [x_1, x_2, ..., x_L] \in \mathbb{R}^{L \times C}$ to obtain $\mathbf{X}_{norm} = [x'_1, x'_2, ..., x'_L] \in \mathbb{R}^{L \times C}$:

$$\mu_{\mathbf{x}} = \frac{1}{L} \sum_{i=1}^{L} x_i, \ \sigma_{\mathbf{x}}^2 = \frac{1}{L} \sum_{i=1}^{L} (x_i - \mu_{\mathbf{x}})^2, \ x'_i = \frac{1}{\sigma_{\mathbf{x}}} \odot (x_i - \mu_{\mathbf{x}}), \tag{9}$$

where $\mu_{\mathbf{x}}, \sigma_{\mathbf{x}} \in \mathbb{R}^{C \times 1}$ are the mean and variance, respectively, $\frac{1}{\sigma_{\mathbf{x}}}$ means the element-wise division and $\odot$ is the element-wise product. Normalization reduces the disparity in distribution among individual input time series, thereby stabilizing the model input distribution.

Then, to remove the trend part from the time series and only preserve the seasonal part for Peri-midFormer, we refer to [12] for time series decomposition, as shown in the following equation:

$$\begin{aligned} \mathbf{X}_t &= AvgPool(Padding(\mathbf{X}_{norm})), \\ \mathbf{X}_s &= \mathbf{X}_{norm} - \mathbf{X}_t, \end{aligned} \tag{10}$$

where $\mathbf{X}_s, \mathbf{X}_t \in \mathbb{R}^{L \times C}$ denote the seasonal and the trend part respectively. We adopt the $AvgPool(\cdot)$ for moving average with the padding operation to keep the series length unchanged.

The seasonal part $\mathbf{X}_s$ obtained after decomposition can be directly input into Peri-midFormer. After the output from Peri-midFormer, we add the trend part back, then de-normalize it to obtain the final output $\hat{\mathbf{Y}} = [\hat{y_1}, \hat{y_2}, ..., \hat{y_T}]$:

$$\mathbf{Y}_s = \mathcal{H}(\mathbf{X}_s), \mathbf{Y} = \mathbf{Y}_s + Projection(\mathbf{X}_t), \hat{y}_i = \sigma_{\mathbf{x}} \odot y_i + \mu_{\mathbf{x}}, \tag{11}$$

where $\mathcal{H}$ represents the Peri-midFormer model, $\mathbf{Y}_s$ represents the output of Peri-midFormer, $Projection(\cdot)$ represents mapping the trend part to the target output length, and $\hat{y}_i = \sigma_{\mathbf{x}} \odot y_i + \mu_{\mathbf{x}}$ denotes de-normalization.

## B    Visualization

To provide a clearer demonstration of Peri-midFormer's representational capabilities, Figure 9, 10 and 11 visualize some of the results for imputation, long-term forecasting and short-term forecasting, respectively. It illustrates that Peri-midFormer outperforms other methods in capturing periodic variations in time series.

(1) Imputation on Weather dataset under 50% mask ratio

Peri-midFormer (ours)

GPT4TS

PatchTST

TimesNet

FEDformer

Dlinear

(2) Imputation on Electricity dataset under 50% mask ratio

Peri-midFormer (ours)

GPT4TS

PatchTST

TimesNet

FEDformer

Dlinear

Figure 9: Visualization of imputation.

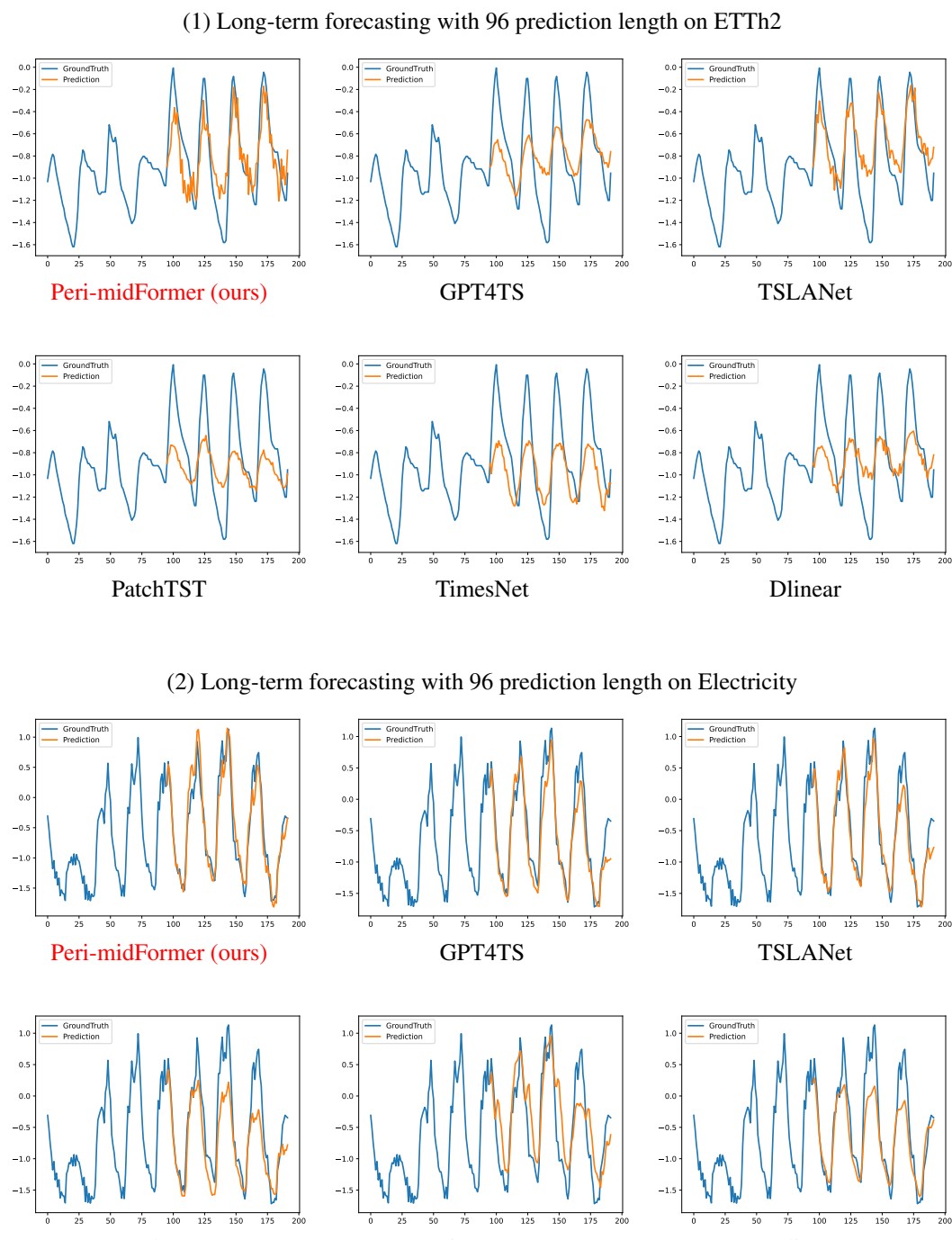

Figure 10: Visualization of long-term forecasting.

(1) Short-term forecasting on M4 Weekly

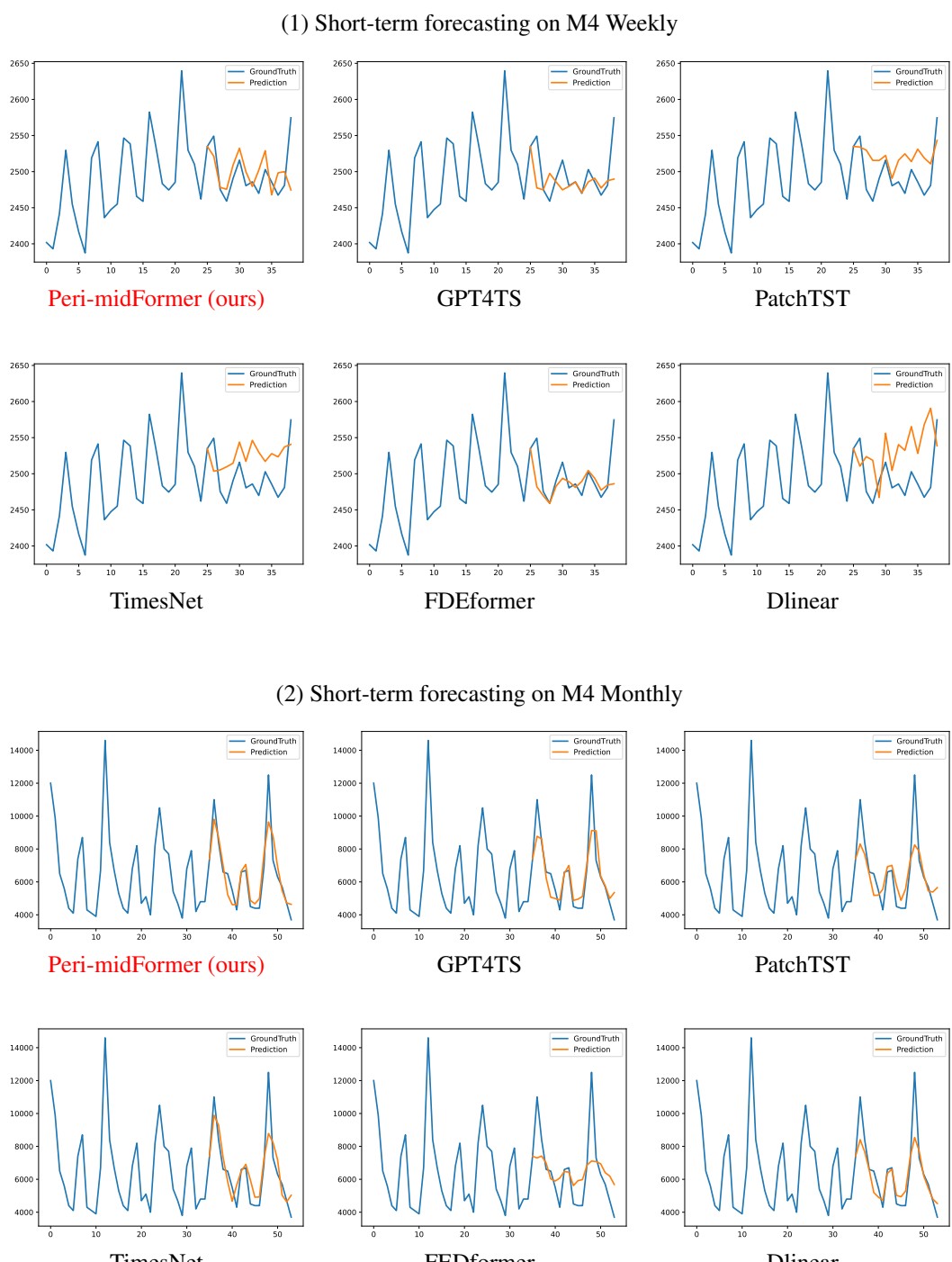

Figure 11: Visualization of short-term forecasting.

## C   Dataset Details

A detailed description of the dataset is given in Table 6.

Table 6: Dataset descriptions. The dataset size is organized in (Train, Validation, Test).

| Tasks | Dataset | Dim | Series Length | Dataset Size | Information (Frequency) |
|---|---|---|---|---|---|
| Forecasting (Long-term) | ETTm1, ETTm2 | 7 | {96, 192, 336, 720} | (34465, 11521, 11521) | Electricity (15 mins) |
| | ETTh1, ETTh2 | 7 | {96, 192, 336, 720} | (8545, 2881, 2881) | Electricity (15 mins) |
| | Electricity | 321 | {96, 192, 336, 720} | (18317, 2633, 5261) | Electricity (Hourly) |
| | Traffic | 862 | {96, 192, 336, 720} | (12185, 1757, 3509) | Transportation (Hourly) |
| | Weather | 21 | {96, 192, 336, 720} | (36792, 5271, 10540) | Weather (10 mins) |
| | Exchange | 8 | {96, 192, 336, 720} | (5120, 665, 1422) | Exchange rate (Daily) |
| Forecasting (short-term) | M4-Yearly | 1 | 6 | (23000, 0, 23000) | Demographic |
| | M4-Quarterly | 1 | 8 | (24000, 0, 24000) | Finance |
| | M4-Monthly | 1 | 18 | (48000, 0, 48000) | Industry |
| | M4-Weakly | 1 | 13 | (359, 0, 359) | Macro |
| | M4-Daily | 1 | 14 | (4227, 0, 4227) | Micro |
| | M4-Hourly | 1 | 48 | (414, 0, 414) | Other |
| Imputation | ETTm1, ETTm2 | 7 | 96 | (34465, 11521, 11521) | Electricity (15 mins) |
| | ETTh1, ETTh2 | 7 | 96 | (8545, 2881, 2881) | Electricity (15 mins) |
| | Electricity | 321 | 96 | (18317, 2633, 5261) | Electricity (15 mins) |
| | Weather | 21 | 96 | (36792, 5271, 10540) | Weather (10 mins) |
| Classification (UEA) | EthanolConcentration | 3 | 1751 | (261, 0, 263) | Alcohol Industry |
| | FaceDetection | 144 | 62 | (5890, 0, 3524) | Face (250Hz) |
| | Handwriting | 3 | 152 | (150, 0, 850) | Handwriting |
| | Heartbeat | 61 | 405 | (204, 0, 205) | Heart Beat |
| | JapaneseVowels | 12 | 29 | (270, 0, 370) | Voice |
| | PEMS-SF | 963 | 144 | (267, 0, 173) | Transportation (Daily) |
| | SelfRegulationSCP1 | 6 | 896 | (268, 0, 293) | Health (256Hz) |
| | SelfRegulationSCP2 | 7 | 1152 | (200, 0, 180) | Health (256Hz) |
| | SpokenArabicDigits | 13 | 93 | (6599, 0, 2199) | Voice (11025Hz) |
| | UWaveGestureLibrary | 3 | 315 | (120, 0, 320) | Gesture |
| Anomaly Detection | SMD | 38 | 100 | (566724, 141681, 708420) | Server Machine |
| | MSL | 55 | 100 | (44653, 11664, 73729) | Spacecraft |
| | SMAP | 25 | 100 | (108146, 27037, 427617) | Spacecraft |
| | SWaT | 51 | 100 | (396000, 99000, 449919) | Infrastructure |
| | PSM | 25 | 100 | (105984, 26497, 87841) | Server Machine |

Since our datasets setup is the same as Timesnet [13], an excerpt from it describes the datasets.

## D   Experimental Details

All the deep learning networks are implemented in PyTorch and trained on NVIDIA 4090 24GB GPU. We repeated each experiment three times to eliminate randomness. The detailed experiment configuration is shown in Table 7.

### D.1   Metrics

We utilize various metrics to evaluate different tasks. For long-term forecasting and imputations, we employ the mean square error (MSE) and mean absolute error (MAE). In anomaly detection, we utilize the F1-score, which combines precision and recall. For short-term forecasting, we utilize the symmetric mean absolute percentage error (SMAPE), mean absolute scaled error (MASE), and overall weighted average (OWA), with OWA being a unique metric used in the M4 competition.

Table 7: Experiment configuration of Peri-midFormer. All the experiments use the ADAM [49] optimizer with the default hyperparameter configuration for $(\beta_1, \beta_2)$ as (0.9, 0.999).

| Tasks / Configurations | Model Hyper-parameter | | | Training Process | | | |
|---|---|---|---|---|---|---|---|
| | $k$ | Layers | $d_{\text{model}}$ | LR* | Loss | Batch Size | Epochs |
| Long-term Forecasting | 2-5 | 1-3 | 128-768 | $10^{-4}$ - $5 \times 10^{-4}$ | MSE | 2-32 | 15 |
| Short-term Forecasting | 2-3 | 1-2 | 256 | $10^{-4}$ | SMAPE | 2 | 15 |
| Imputation | 2-5 | 1-2 | 64-768 | $10^{-4}$ - $5 \times 10^{-4}$ | MSE | 2-4 | 15 |
| Classification | 2-8 | 1-4 | 16-128 | $10^{-4}$ - $5 \times 10^{-3}$ | Cross Entropy | 2-64 | 20 |
| Anomaly Detection | 2-5 | 1-4 | 8-32 | $10^{-4}$ - $5 \times 10^{-3}$ | MSE | 2-32 | 3 |

∗ LR means the initial learning rate.

These metrics are computed as follows:

$$
\begin{aligned}
\text{SMAPE} &= \frac{200}{T} \sum_{i=1}^{T} \frac{|\mathbf{X}_i - \widehat{\mathbf{Y}}_i|}{|\mathbf{X}_i| + |\widehat{\mathbf{Y}}_i|}, \\
\text{MAPE} &= \frac{100}{T} \sum_{i=1}^{T} \frac{|\mathbf{X}_i - \widehat{\mathbf{Y}}_i|}{|\mathbf{X}_i|}, \\
\text{MASE} &= \frac{1}{T} \sum_{i=1}^{T} \frac{|\mathbf{X}_i - \widehat{\mathbf{Y}}_i|}{\frac{1}{T-q} \sum_{j=q+1}^{T} |\mathbf{X}_j - \mathbf{X}_{j-q}|}, \\
\text{OWA} &= \frac{1}{2} \left[ \frac{\text{SMAPE}}{\text{SMAPE}_{\text{Naïve2}}} + \frac{\text{MASE}}{\text{MASE}_{\text{Naïve2}}} \right],
\end{aligned}
\tag{12}
$$

where $q$ is the periodicity of the data. $\mathbf{X}, \widehat{\mathbf{Y}} \in \mathbb{R}^{T \times C}$ are the ground truth and prediction result of the future with $T$ time pints and $C$ dimensions. $\mathbf{X}_i$ means the $i$-th future time point.

# E   Model Analysis

## E.1   Hyper Parameter Analysis and Model Limitations

In Equation (1), we introduced a hyperparameter $k$ to select the most important frequency, which also determines the number of levels in the Periodic Pyramid. We conducted sensitivity analysis on it, as shown in Figure 12. It's evident that our proposed Peri-midFormer exhibits relatively stable performance across different choices of $k$ for all four tasks. However, there are still some fluctuation in results among different $k$ values depending on the task and dataset, which are determined by the periodic characteristics in the dataset. To illustrate this, we visualize individual data for long-term forecasting and classification tasks, as shown in Figure 13. It can be observed that in the long-term forecasting task, the Etth1 dataset exhibits clear periodicity. Therefore, with larger $k$, Peri-midFormer can capture more periodic information, resulting in better performance. In contrast, the Etth2 dataset has less obvious periodicity, so it achieves better results with smaller $k$, as larger $k$ introduce unnecessary noise, affecting model performance. In the classification task, the EthanolConcentration dataset is difficult to classify, whereas a larger $k$ allows for the construction of Periodic Pyramid with more levels, thus extracting more representative features and achieving higher accuracy. In addition, the SelfRegulationSCP1 dataset contains a lot of high-frequency noise, so larger $k$ would focus on irrelevant information, leading to decreased accuracy. Therefore, we adjusted $k$ values differently for different tasks and datasets, as shown in the range in Table 7.

The above analysis reveals that the limitation of Peri-midFormer lies in its inability to fully leverage its advantages on datasets with poor periodicity characteristics. We plan to address this issue in our future work.

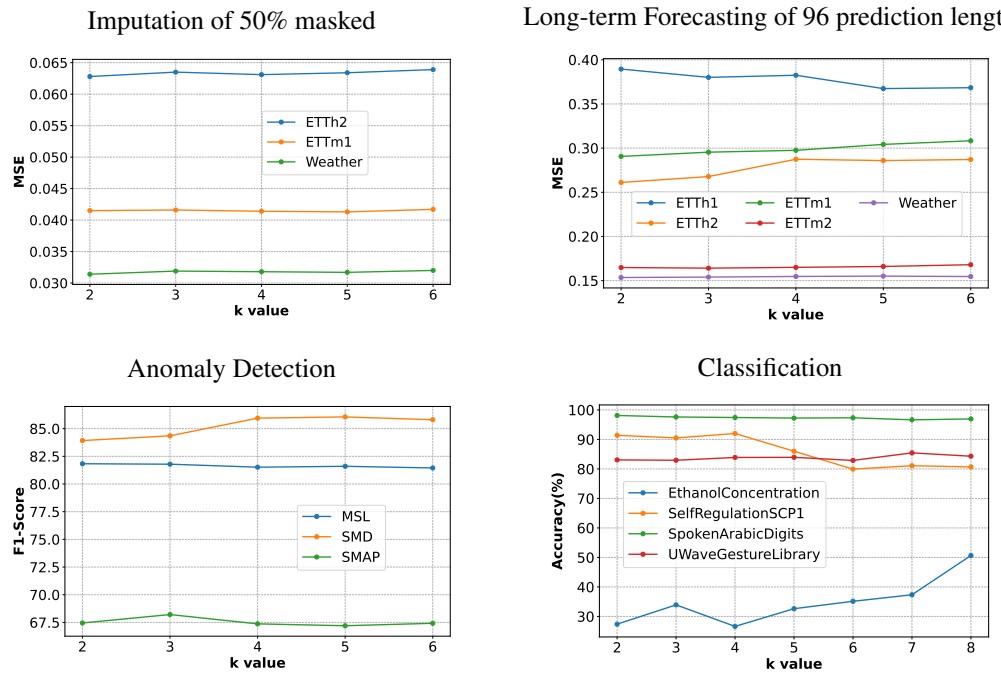

Figure 12: Sensitivity analysis of hyper-parameters $k$ in each task.

(1) Long-term Forecasting task

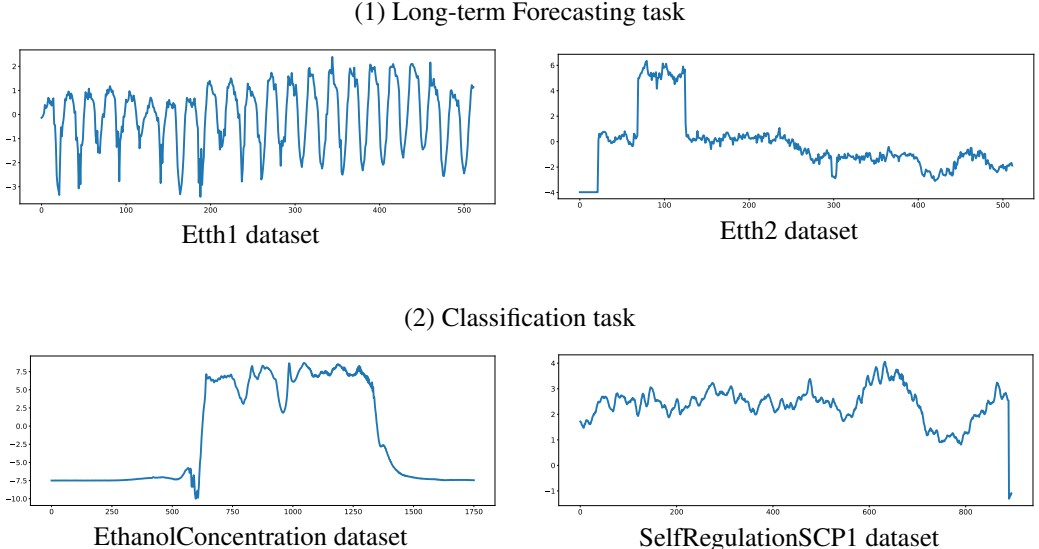

Figure 13: Visualization of the Etth1 and Etth2 datasets for the Long-term forecasting task (1), and the EthanolConcentration and SelfRegulationSCP1 datasets for the classification task (2).

### E.2 Periodic Pyramid Attention Mechanism Analysis

To illustrate the PPAM more clearly, we visualize the original time series and attention scores within the periodic pyramid for the SelfRegulationSCP2 dataset in the classification task, as shown in Figure 14. It is evident that the attention scores among components are distributed based on inclusion and adjacency relationships, meaning that components in different levels with inclusion relationships or those in the same level have higher attention scores. This demonstrates the rationality of the Periodic Pyramid structure. Additionally, when the hyperparameter $k$ in Equation (1) is set to 2, the PPAM degenerates into periodic full attention, wherein attention is computed among all periodic

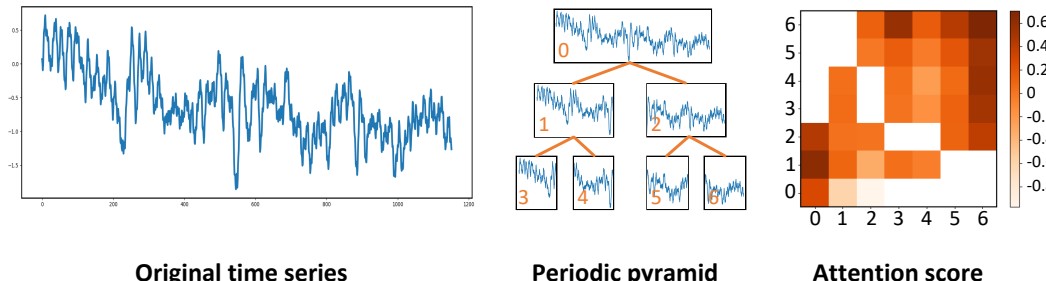

**Original time series**      **Periodic pyramid**      **Attention score**

Figure 14: Visualization of the original time series and attention scores within the periodic pyramid for the SelfRegulationSCP2 dataset in the classification task. The left side shows the original data, the middle displays the corresponding pyramid structure, and the right side depicts the attention scores within the pyramid. In this example, $k = 3$, $f_1 = 1$, $f_2 = 2$, $f_3 = 4$.

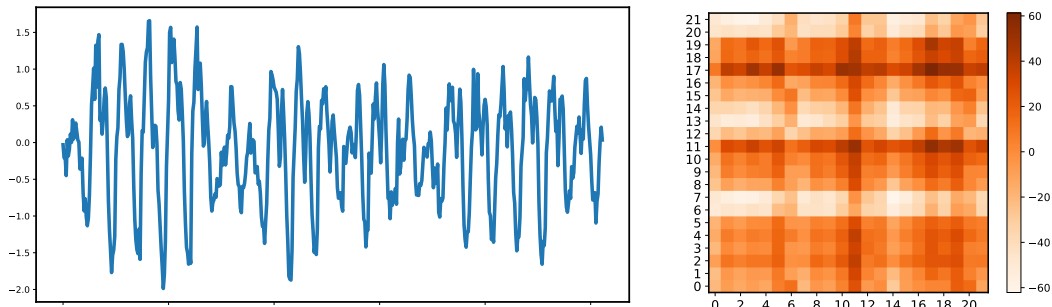

Figure 15: Original time series of Electricity dataset (left) and Periodic Pyramid Attention score (right).

components. To illustrate this, we visualize the Electricity dataset in the long-term forecasting task and its corresponding attention distribution, as shown in Figure 15. The figure shows that the Periodic Attention Mechanism can capture the dependencies among the periodic components and identify which components belong to a longer component (note that $k = 2$ corresponds to 21 periodic components with larger amplitudes). This is attributed to the separation of the periodic components, which explicitly expresses the hidden periodic inclusion relationships in the time series. The above analysis demonstrates the effectiveness of the PPAM.

### E.3 Periodic Feature Flows Analysis

To intuitively understand the Periodic Feature Flows, we visualize it as shown in Figure 16. On the left side is the pyramid representation of the Periodic Feature Flows, with the horizontal axis representing the number of feature flows and the vertical axis representing the length of each feature flow. The feature flows are divided into multiple levels, each containing multiple periodic components. The red box encloses one feature flow, with its position from top to bottom corresponding to the pyramid from top to bottom. It can be observed that some adjacent feature flows are the same at the corresponding positions, this is because they pass through the same periodic component. On the right side, the waveform of each feature flow is displayed in different colors, with the position from left to right corresponding to the top to the bottom of the pyramid. It can be observed that each feature flow differs, which is why it is necessary to aggregate different feature flows. The aim is to fully utilize the information from each periodic component to better reconstruct the target sample.

### E.4 Training/Inferencing Cost

To further validate the computational complexity and scalability of the proposed method, we conducted detailed experiments on Electricity and ETTh1 datasets for the long-term forecasting task. These experiments focused on the complexity and actual time of training and inference, as well as memory usage, with results presented in Tables 8 and 9. It can be seen that our proposed Peri-

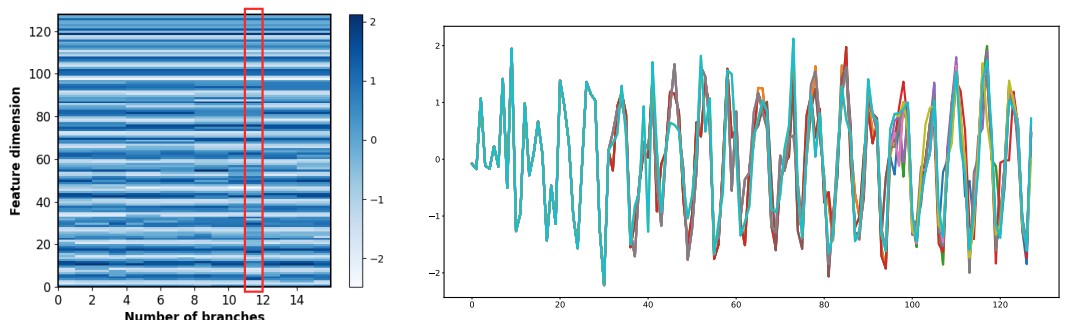

Figure 16: Visualization of the pyramid form of Periodic Feature Flows (left) and the waveform of Periodic Feature Flows (right).

Table 8: Complexity and scalability experiments in the long-term forecasting of length 720 on Electricity

| Methods | Train FLOPs | Test FLOPs | GPU memory | Train CPU memory | Test CPU memory | Train time (s) | Test time (per sample) | MSE |
|---|---|---|---|---|---|---|---|---|
| **Peri-midFormer** | 244G | 61G | 4816M | 2753M | 2101M | 5807s | 0.0263s | 0.181 |
| Time-LLM | 12.69T | - | 61454M | 12503M | - | - | - | 0.192 |
| GPT4TS | 716G | 3G | 11236M | 10572M | 10471M | 22366s | 0.0280s | 0.294 |
| PatchTST | 549G | 137G | 11500M | 8894M | 2717M | 7029s | 0.0219s | 0.294 |
| TimesNet | 29.6T | 934G | 18112M | 21021M | 4225M | 9769s | 0.0501s | 0.294 |
| DLinear | 8G | 237M | 1892M | 8045M | 1427M | 331s | 0.0032s | 0.204 |
| Autoformer | 238G | 8G | 13157M | 8488M | 2738M | 1477s | 0.1390s | 0.236 |

Table 9: Complexity and scalability experiments in the long-term forecasting of length 720 on ETTh1

| Methods | Train FLOPs | Test FLOPs | GPU memory | Train CPU memory | Test CPU memory | Train time (s) | Test time (per sample) | MSE |
|---|---|---|---|---|---|---|---|---|
| **Peri-midFormer** | 23.51G | 4.28G | 2443M | 3190M | 75M | 1623s | 0.0036s | 0.445 |
| Time-LLM | 25.36T | 25.36T | 94994M | 5795M | 2893M | 30506s | 0.0384s | 0.442 |
| GPT4TS | 716G | 3G | 11657M | 3250M | 170M | 2920s | 0.0043s | 0.477 |
| PatchTST | 51.24G | 1.58G | 3683M | 2895M | 30M | 550s | 0.0013s | 0.473 |
| TimesNet | 116.32G | 3.67G | 2932M | 3005M | 20M | 1640s | 0.0076s | 0.551 |
| DLinear | 165.05M | 5.17M | 1449M | 1310M | 28M | 190s | 0.0006s | 0.472 |
| Autoformer | 202.34G | 6.34G | 12981M | 3216M | 16M | 3400s | 0.0123s | 0.501 |

midFormer demonstrates a significant advantage in computational complexity on the Electricity dataset and achieves the lowest MSE. Similarly, on the ETTh1 dataset, Peri-midFormer's computational cost and inference time are not disadvantages. Instead, it achieves a lower MSE, second only to Time-LLM, while having much lower computational cost and inference time compared to it. These results highlight the advantages of our method in terms of computational complexity and scalability.

### E.5 Pre-interpolation

Since Peri-midFormer is designed to focus on the periodic components of time series, directly handling data with missing values in imputation tasks may prevent it from correctly capturing the periodic characteristics of the original data without missing values, thus affecting its imputation effectiveness. Therefore, to adapt Peri-midFormer for imputation tasks, we first apply a linear interpolation strategy (Equation (13)) to the data with missing values to partially restore the periodic characteristics of the original data before inputting it into Peri-midFormer for further imputation. The visualization of original data, data with 50% missing values, and pre-interpolated data of Electricity dataset are illustrated in Figure 17. It is evident that missing values significantly disrupt the periodic

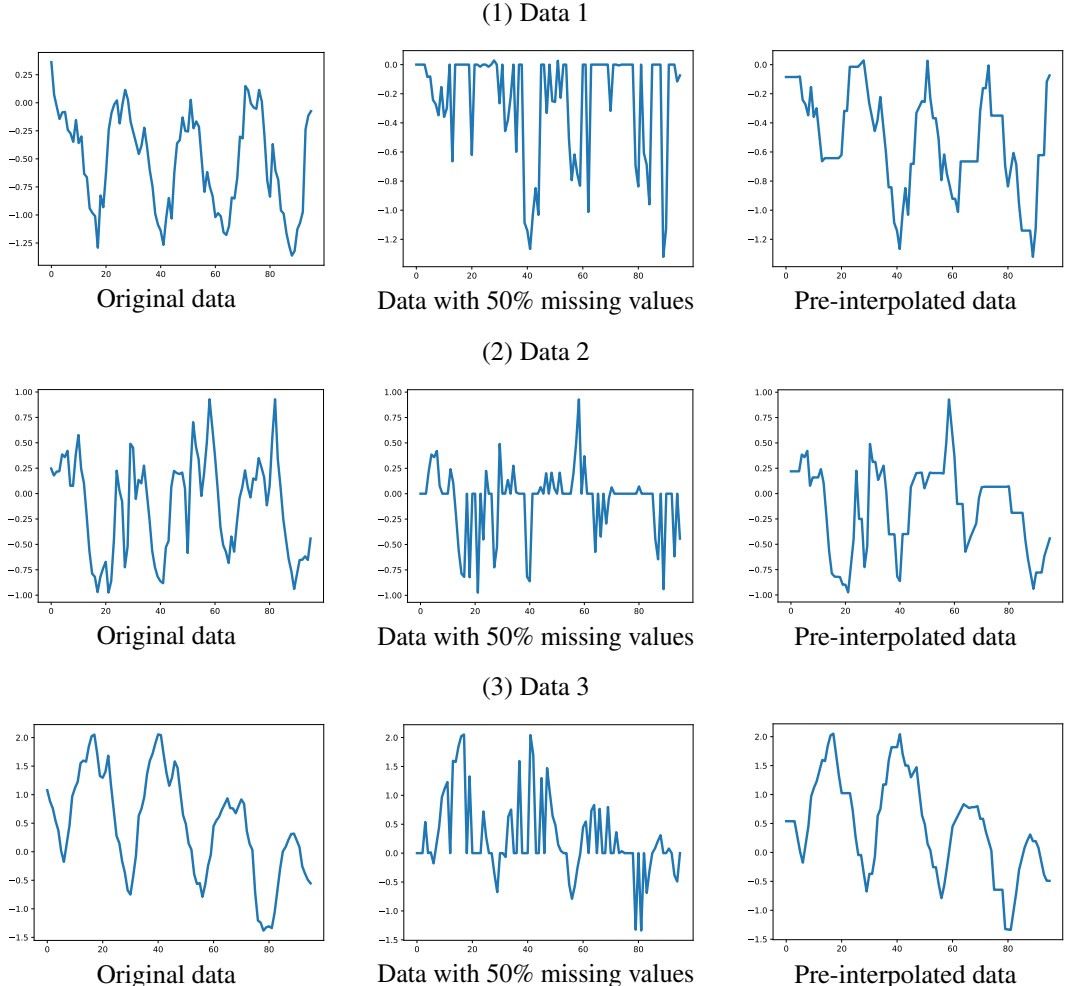

Figure 17: Visualization of original data, data with 50% missing values, and pre-interpolated data of Electricity dataset.

characteristics of the original data, while pre-interpolation partially restores them. It is worth noting that this is not a speculative action but an effort to further explore the potential of deep learning models in imputation tasks. Our goal is for deep models to capture subtle variations in time series to achieve imputation in the details rather than wasting effort on goals achievable through simple linear interpolation. Additionally, since we can retain the indices of missing values in practical applications, there is no concern about the reliability of evaluating the imputation results. To validate the effectiveness of the pre-interpolation strategy, we applied it to other methods, as shown in Table 10. We conducted experiments on four mask ratios $\{0.125, 0.25, 0.375, 0.5\}$, where the first row uses only the pre-interpolation strategy. From the results, it can be seen that pre-interpolation significantly improves the performance of each method across all four mask ratios, demonstrating its importance for deep learning-based methods in interpolation tasks.

The simple linear interpolation strategy we adopted can be represented by the equation:

$$x_{inter} = \begin{cases} \frac{x_{before}+x_{after}}{2}, & if \ (x_{before} \neq \text{None}) \ and \ (x_{after} \neq \text{None}) \\ x_{after}, & if \ (x_{before} = \text{None}) \ and \ (x_{after} \neq \text{None}) \ , \\ x_{before}, & if \ (x_{before} \neq \text{None}) \ and \ (x_{after} = \text{None}) \end{cases} \tag{13}$$

where $x_{inter}$ represents the value used to replace the 0 value, $x_{before}$ represents the nearest non-zero value before the 0 value, and $x_{after}$ represents the nearest non-zero value after the 0 value.

Table 10: Ablation Experiments of pre-interpolation in inputation task on ECL dataset.

| Methods | Metrics | w/o pre-interpolation | | | | Pre-interpolation | | | |
|---|---|---|---|---|---|---|---|---|---|
| | | **0.125** | **0.25** | **0.375** | **0.5** | **0.125** | **0.25** | **0.375** | **0.5** |
| Only pre-interpolation | MSE | - | - | - | - | 0.086 | 0.110 | 0.149 | 0.206 |
| | MAE | - | - | - | - | 0.188 | 0.213 | 0.251 | 0.301 |
| **Pyri-midFormer** | MSE | 0.073 | 0.092 | 0.107 | 0.122 | 0.044 | 0.052 | 0.063 | 0.079 |
| | MAE | 0.187 | 0.214 | 0.231 | 0.248 | 0.135 | 0.149 | 0.166 | 0.189 |
| Timesnet | MSE | 0.085 | 0.089 | 0.094 | 0.100 | 0.081 | 0.083 | 0.086 | 0.091 |
| | MAE | 0.202 | 0.206 | 0.213 | 0.221 | 0.196 | 0.198 | 0.201 | 0.207 |
| Pyraformer | MSE | 0.297 | 0.294 | 0.296 | 0.299 | 0.165 | 0.165 | 0.171 | 0.173 |
| | MAE | 0.383 | 0.380 | 0.381 | 0.383 | 0.290 | 0.291 | 0.293 | 0.295 |
| Dlinear | MSE | 0.092 | 0.118 | 0.144 | 0.175 | 0.050 | 0.062 | 0.789 | 0.105 |
| | MAE | 0.214 | 0.247 | 0.276 | 0.305 | 0.144 | 0.164 | 0.189 | 0.225 |
| PatchTST | MSE | 0.055 | 0.065 | 0.076 | 0.091 | 0.050 | 0.059 | 0.070 | 0.087 |
| | MAE | 0.160 | 0.175 | 0.189 | 0.208 | 0.148 | 0.164 | 0.181 | 0.202 |
| ETSformer | MSE | 0.196 | 0.207 | 0.219 | 0.235 | 0.102 | 0.104 | 0.109 | 0.117 |
| | MAE | 0.321 | 0.332 | 0.344 | 0.357 | 0.230 | 0.233 | 0.238 | 0.247 |

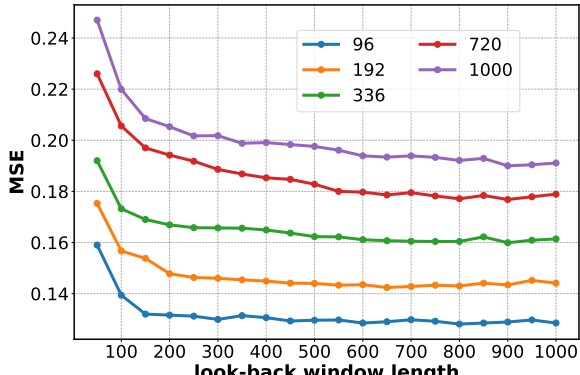

Figure 18: Performance of different length look-back window on the long-term forecasting task in Electricity dataset. Prediction lengths are $\{96, 192, 336, 720, 1000\}$.

## E.6 Look-back window length

Here, we investigate the performance of Peri-midFormer under different look-back window lengths in long-term forecasting task on the Electricity dataset. As shown in Figure 18, we experimented not only with prediction lengths of $\{96, 192, 336, 720\}$, but also with longer lengths such as 1000. It can be observed that there is a clear improvement in performance across all five prediction lengths as the look-back window length increases. This improvement is particularly notable for longer prediction lengths such as 336, 720, and 1000. It indicates that Peri-midFormer can effectively utilize more historical information, as longer look-back windows contain more stable periodic characteristics, which is precisely what Peri-midFormer requires.

## E.7 Time Series Decomposition Analysis

Peri-midFormer performs well on the Exchange dataset for long-term forecasting task and the Yearly dataset for short-term forecasting. Although these two datasets lack obvious periodicity, they exhibit strong trends, as shown in Figure (19)). Peri-midFormer uses a time series decomposition strategy, where the trend part is first separated from the original data before dividing the periodic components. After the output of Peri-midFormer, the predicted trend part is added back, as illustrated in Figure (8). The separated trend part is easier to predict, especially when the trend is strong. This is why Peri-midFormer achieves strong performance on the Exchange and Yearly datasets.

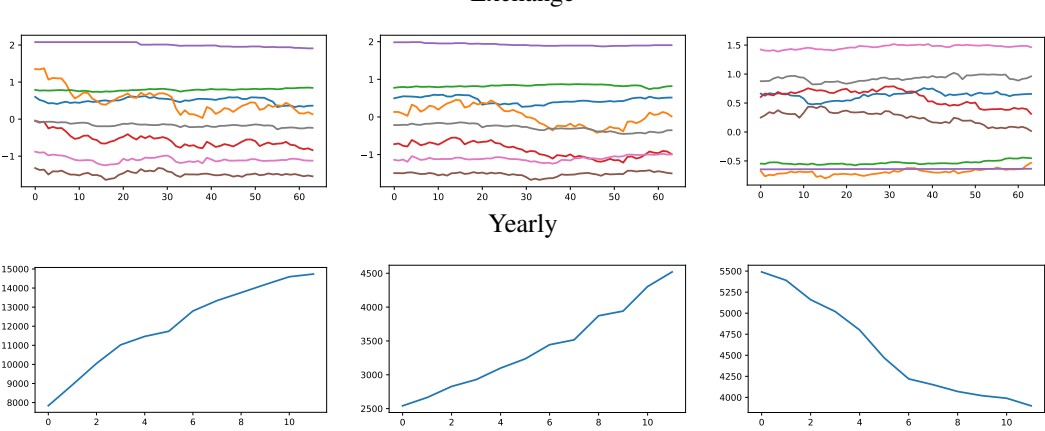

Figure 19: Visualization of Exchange and Yearly dataset.

# F Proof

To demonstrate the essence of attention computation among multi-level periodic components, we need to analyze how the interactions between periodic components at different levels affect the final feature extraction. In time series analysis, different periodic components correspond to different time scales. This means that through decomposition, we can capture components of various frequencies within the time series. The essence of the periodic pyramid is to capture these different frequency components through its hierarchical structure.

Using single-channel data as an example, and given that we adopt an independent channel strategy, this can be easily extended to all channels. Assume the time series $x(t)$ can be decomposed into multiple periodic components $x_n(t)$ :

$$x(t) = \sum_{n=1}^{N} x_n(t). \tag{14}$$

Taking two different periodic components as examples:

$$x_i(t) = A_i \sin\left(\frac{2\pi t}{T_i} + \phi_i\right), x_j(t) = A_j \cos\left(\frac{2\pi t}{T_j} + \phi_j\right), \tag{15}$$

where $A$ is amplitude, $T$ is period, and $\phi$ is phase. Due to the overlap and inclusion relationships between different periodic components, we employ an attention mechanism in the periodic pyramid to capture the similarities between different periodic components, focusing on important periodic features. When applying the attention mechanism, we have:

$$Q_i = W_Q x_i(t), \quad K_j = W_K x_j(t), \quad V_j = W_V x_j(t), \tag{16}$$

where $W_Q$, $W_K$ and $W_V$ are learnable weight matrices. From Equations (15) and (16):

$$Q_i = W_Q A_i \sin\left(\frac{2\pi t}{T_i} + \phi_i\right), \quad K_j = W_K A_j \cos\left(\frac{2\pi t}{T_j} + \phi_j\right). \tag{17}$$

Further, the dot-product attention can be expressed as:

$$Q_i K_j^T = A_i A_j \left(W_Q \sin\left(\frac{2\pi t}{T_i} + \phi_i\right)\right) \left(W_K \cos\left(\frac{2\pi t}{T_j} + \phi_j\right)\right)^T. \tag{18}$$

Using the trigonometric identity $\sin(a)\cos(b) = \frac{1}{2}[\sin(a+b) + \sin(a-b)]$, the dot-product $Q_i K_j^T$ can be further expressed as:

$$Q_i K_j^T = \frac{1}{2} A_i A_j \left\{ W_Q \left[ \sin\left(\frac{2\pi t}{T_i} + \phi_i + \frac{2\pi t}{T_j} + \phi_j\right) + \sin\left(\frac{2\pi t}{T_i} + \phi_i - \frac{2\pi t}{T_j} - \phi_j\right) \right] \right\} (W_K)^T. \tag{19}$$

Based on this, considering the periodicity and symmetry of $\sin(a + b)$ and $\sin(a - b)$, when the periods of two time series components are close / same (intra-layer attention in the pyramid, see the right side of Figure 3 in the original paper) or have overlapping / inclusive parts (inter-layer attention in the pyramid, see the right side of Figure 3 in the original paper), the values of these two sine functions will be highly correlated, resulting in a large $Q_i K_j^T$ value. This indicates that the periodic pyramid model can effectively capture similar periodic patterns across different time scales.

Next, incorporating this into the calculation of the attention score:

$$s_{ij} = \frac{\exp\left(\frac{Q_i K_j^T}{\sqrt{d_k}}\right)}{\sum\limits_{j'} \exp\left(\frac{Q_i K_{j'}^T}{\sqrt{d_k}}\right)}. \tag{20}$$

It can be seen that the attention scores between highly correlated periodic components will be higher, which we have already validated in Figures 13 and 14 of the original paper.

Further, the attention vector $\mathbf{a}_i$ of $x_i(t)$ can be obtained as:

$$\mathbf{a}_i = \sum_j s_{ij} V_j , \tag{21}$$

where $V_j = W_V x_j(t) = W_V A_j \cos\left(\frac{2\pi t}{T_j} + \phi_j\right)$.

From the above derivation, it can be seen that the attention mechanism can measure the similarity between different periodic components. This similarity reflects the alignment between different periodic components in the time series, allowing the model to capture important periodic patterns. By capturing these periodic patterns, the periodic pyramid can extract key features of the time series, resulting in a comprehensive and accurate time series representation. This representation not only includes information across different time scales but also enhances the representation of important periodic patterns.

## G   Broader Impacts

Our research has implications for time-series-based analyses such as weather forecasting and anomaly detection for industrial maintenance, as discussed in the Introduction 1. Our work carries no negative social impact.

## H   Full Results

### H.1   Full Results of Classification (Table 11)

### H.2   Full Results of Short-term Forecasting (Table 12)

### H.3   Full Results of Long-term Forecasting (Table 13 and 14)

### H.4   Full Results of Imputation (Table 15)

### H.5   Full Results of Anomaly Detection (Table 16)

Table 11: Full results for the classification task. (∗ means former, T-LLM means Time-LLM, GPT means GPT4TS, T-Net means TimesNet, Patch means PatchTST, Light means LightTS, Station means the Non-stationary Transformer.) We report the classification accuracy (%) as the result. The standard deviation is within 1%. We reproduced the results of PatchTST by https://github.com/thuml/Time-Series-Library, reproduced TSLANet by https://github.com/emadeldeen24/TSLANet, and copied the others from GPT4TS [20]. **Red**: best, Blue: second best.

| Datasets / Models | Rocket [33] | Patch [19] | TCN [8] | Trans∗ [48] | Re∗ [28] | Pyra∗ [10] | Auto∗ [12] | Station [11] | FED∗ [17] | ETS∗ [29] | Flow∗ [35] | DLinear [14] | Light [15] | T-Net [13] | GPT [20] | TLSANet [27] | Peri-mid Former |
|---|---|---|---|---|---|---|---|---|---|---|---|---|---|---|---|---|---|
| EthanolConcentration | 45.2 | 29.6 | 28.9 | 32.7 | 31.9 | 30.8 | 31.6 | 32.7 | 31.2 | 28.1 | 33.8 | 32.6 | 29.7 | 35.7 | 34.2 | 30.4 | **50.6** |
| FaceDetection | 64.7 | 67.8 | 52.8 | 67.3 | 68.6 | 65.7 | 68.4 | 68.0 | 66.0 | 66.3 | 67.6 | 68.0 | 67.5 | 68.6 | **69.2** | 66.7 | 68.7 |
| Handwriting | **58.8** | 23.2 | 53.3 | 32.0 | 27.4 | 29.4 | 36.7 | 31.6 | 28.0 | 32.5 | 33.8 | 27.0 | 26.1 | 32.1 | 32.7 | 57.88 | 31.5 |
| Heartbeat | 75.6 | 75.7 | 75.6 | 76.1 | 77.1 | 75.6 | 74.6 | 73.7 | 73.7 | 71.2 | 77.6 | 75.1 | 75.1 | 78.0 | 77.2 | 77.5 | **95.1** |
| JapaneseVowels | 96.2 | 94.0 | 98.9 | 98.7 | 97.8 | 98.4 | 96.2 | **99.2** | 98.4 | 95.9 | 98.9 | 96.2 | 96.2 | 98.4 | 98.6 | 99.2 | 97.3 |
| PEMS-SF | 75.1 | 80.9 | 68.6 | 82.1 | 82.7 | 83.2 | 82.7 | 87.3 | 80.9 | 86.0 | 83.8 | 75.1 | 88.4 | **89.6** | 87.9 | 83.8 | 88.2 |
| SelfRegulationSCP1 | 90.8 | 82.2 | 84.6 | 92.2 | 90.4 | 88.1 | 84.0 | 89.4 | 88.7 | 89.6 | 92.5 | 87.3 | 89.8 | 91.8 | **93.2** | 91.8 | 92.1 |
| SelfRegulationSCP2 | 53.3 | 53.6 | 55.6 | 53.9 | 56.7 | 53.3 | 50.6 | 57.2 | 54.4 | 55.0 | 56.1 | 50.5 | 51.1 | 57.2 | 59.4 | **61.6** | 58.2 |
| SpokenArabicDigits | 71.2 | 98.0 | 95.6 | 98.4 | 97.0 | 99.6 | **100.0** | **100.0** | **100.0** | **100.0** | 98.8 | 81.4 | **100.0** | 99.0 | 99.2 | 99.9 | 98.2 |
| UWaveGestureLibrary | **94.4** | 81.7 | 88.4 | 85.6 | 85.6 | 83.4 | 85.9 | 87.5 | 85.3 | 85.0 | 86.6 | 82.1 | 80.3 | 85.3 | 88.1 | 91.2 | 85.6 |
| Average Accuracy | 72.5 | 68.7 | 70.3 | 71.9 | 71.5 | 70.8 | 71.1 | 72.7 | 70.7 | 71.0 | 73.0 | 67.5 | 70.4 | 73.6 | 74.0 | 76.0 | **76.5** |

Table 12: Full results for the short-term forecasting task in the M4 dataset. (∗ means former, T-LLM means Time-LLM, GPT means GPT4TS, T-Net means TimesNet, Patch means PatchTST, HiTS means N-HiTS, BEATS means N-BEATS, Light means LightTS, Station means the Non-stationary Transformer.) The standard deviation is within 0.5%. We copied the results of Time-LLM from Time-LLM [25], Pyraformer from TimesNet [13], and the remaining results from GPT4TS [20]. **Red**: best, Blue: second best.

| | Models | Peri-mid Former | T-LLM [25] | GPT [20] | T-Net [13] | Patch [19] | HiTS [30] | BEATS [31] | ETS∗ [29] | Light [15] | DLinear [14] | FED∗ [17] | Station [11] | Auto∗ [12] | Pyra∗ [10] | In∗ [16] | Re∗ [28] |
|---|---|---|---|---|---|---|---|---|---|---|---|---|---|---|---|---|---|
| Yearly | SMAPE | **13.358** | 13.419 | 13.531 | 13.387 | 13.477 | 13.418 | 13.436 | 18.009 | 14.247 | 16.965 | 13.728 | 13.717 | 13.974 | 15.530 | 14.727 | 16.169 |
| | MASE | 3.021 | 3.005 | 3.015 | **2.996** | 3.019 | 3.045 | 3.043 | 4.487 | 3.109 | 4.283 | 3.048 | 3.078 | 3.134 | 3.711 | 3.418 | 3.800 |
| | OWA | 0.789 | 0.789 | 0.793 | **0.786** | 0.792 | 0.793 | 0.794 | 1.115 | 0.827 | 1.058 | 0.803 | 0.807 | 0.822 | 0.942 | 0.881 | 0.973 |
| Quarterly | SMAPE | **10.058** | 10.110 | 10.177 | 10.100 | 10.380 | 10.202 | 10.124 | 13.376 | 11.364 | 12.145 | 10.792 | 10.958 | 11.338 | 15.449 | 11.360 | 13.313 |
| | MASE | 1.170 | 1.178 | 1.194 | 1.182 | 1.233 | 1.194 | **1.169** | 1.906 | 1.328 | 1.520 | 1.283 | 1.325 | 1.365 | 2.350 | 1.401 | 1.775 |
| | OWA | **0.883** | 0.889 | 0.898 | 0.890 | 0.921 | 0.899 | 0.886 | 1.302 | 1.000 | 1.106 | 0.958 | 0.981 | 1.012 | 1.558 | 1.027 | 1.252 |
| Monthly | SMAPE | 12.717 | 12.980 | 12.894 | **12.670** | 12.959 | 12.791 | 12.677 | 14.588 | 14.014 | 13.514 | 14.260 | 13.917 | 13.958 | 17.642 | 14.062 | 20.128 |
| | MASE | **0.933** | 0.963 | 0.956 | **0.933** | 0.970 | 0.969 | 0.937 | 1.368 | 1.053 | 1.037 | 1.102 | 1.097 | 1.103 | 1.913 | 1.141 | 2.614 |
| | OWA | 0.879 | 0.903 | 0.897 | **0.878** | 0.905 | 0.899 | 0.880 | 1.149 | 0.981 | 0.956 | 1.012 | 0.998 | 1.002 | 1.511 | 1.024 | 1.927 |
| Others | SMAPE | 4.845 | **4.795** | 4.940 | 4.891 | 4.952 | 5.061 | 4.925 | 7.267 | 15.880 | 6.709 | 4.954 | 6.302 | 5.485 | 24.786 | 24.460 | 32.491 |
| | MASE | 3.217 | **3.178** | 3.228 | 3.302 | 3.347 | 3.216 | 3.391 | 5.240 | 11.434 | 4.953 | 3.264 | 4.064 | 3.865 | 18.581 | 20.960 | 33.355 |
| | OWA | 1.017 | **1.006** | 1.029 | 1.035 | 1.049 | 1.040 | 1.053 | 1.591 | 3.474 | 1.487 | 1.036 | 1.304 | 1.187 | 5.538 | 5.879 | 8.679 |
| Weighted Average | SMAPE | 11.833 | 11.983 | 11.991 | **11.829** | 12.059 | 11.927 | 11.851 | 14.718 | 13.525 | 13.639 | 12.840 | 12.780 | 12.909 | 16.987 | 14.086 | 18.200 |
| | MASE | **1.584** | 1.595 | 1.600 | 1.585 | 1.623 | 1.613 | 1.599 | 2.408 | 2.111 | 2.095 | 1.701 | 1.756 | 1.771 | 3.265 | 2.718 | 4.223 |
| | OWA | **0.850** | 0.859 | 0.861 | 0.851 | 0.869 | 0.861 | 0.855 | 1.172 | 1.051 | 1.051 | 0.918 | 0.930 | 0.939 | 1.480 | 1.230 | 1.775 |

Table 13: Full results of 512 look-back window length in long-term forecasting task (since FITS defaults to a look-back window length of 720, its results at 720 length are attached at the end). The standard deviation is within 0.5%. We copied the results of GPT4TS from GPT4TS [20], Time-LLM from TSLANet [27], reproduced TSLANet by `https://github.com/emadeldeen24/TSLANet`, and reproduced the others by `https://github.com/thuml/Time-Series-Library`. **Red**: best, Blue: second best.

| Look-back | | 512 | | | | | | | | | | | | | | | | | 720 | |
|---|---|---|---|---|---|---|---|---|---|---|---|---|---|---|---|---|---|---|---|---|---|
| Methods | | **Peri-mid Former** | | GPT4TS [20] | | TSLANet [27] | | Time-LLM [25] | | FITS [21] | | Dlinear [14] | | PatchTST [19] | | TimesNet [13] | | Pyraformer [10] | | FITS [21] | |
| Metrics | | MSE | MAE | MSE | MAE | MSE | MAE | MSE | MAE | MSE | MAE | MSE | MAE | MSE | MAE | MSE | MAE | MSE | MAE | MSE | MAE |
| Weather | 96 | 0.153 | 0.200 | 0.162 | 0.212 | 0.196 | 0.232 | 0.147 | 0.201 | 0.172 | 0.226 | 0.171 | 0.231 | 0.147 | 0.199 | 0.159 | 0.214 | 0.215 | 0.298 | 0.170 | 0.225 |
| | 192 | 0.203 | 0.251 | 0.204 | 0.248 | 0.241 | 0.269 | 0.189 | 0.235 | 0.216 | 0.262 | 0.213 | 0.270 | 0.196 | 0.246 | 0.220 | 0.269 | 0.252 | 0.335 | 0.213 | 0.260 |
| | 336 | 0.255 | 0.294 | 0.254 | 0.286 | 0.297 | 0.308 | 0.262 | 0.279 | 0.261 | 0.295 | 0.259 | 0.311 | 0.243 | 0.283 | 0.276 | 0.307 | 0.282 | 0.352 | 0.258 | 0.295 |
| | 720 | 0.321 | 0.338 | 0.326 | 0.337 | 0.371 | 0.356 | 0.304 | 0.316 | 0.326 | 0.342 | 0.322 | 0.362 | 0.317 | 0.334 | 0.342 | 0.354 | 0.373 | 0.412 | 0.321 | 0.340 |
| | Avg | 0.233 | 0.271 | 0.237 | 0.270 | 0.276 | 0.291 | **0.225** | **0.257** | 0.244 | 0.281 | 0.241 | 0.294 | 0.226 | 0.266 | 0.249 | 0.286 | 0.281 | 0.349 | 0.241 | 0.280 |
| ETTh1 | 96 | 0.367 | 0.396 | 0.376 | 0.397 | 0.374 | 0.402 | 0.362 | 0.392 | 0.372 | 0.396 | 0.367 | 0.397 | 0.374 | 0.402 | 0.442 | 0.457 | 0.702 | 0.628 | 0.379 | 0.402 |
| | 192 | 0.404 | 0.420 | 0.416 | 0.418 | 0.416 | 0.431 | 0.398 | 0.418 | 0.405 | 0.415 | 0.402 | 0.420 | 0.415 | 0.433 | 0.492 | 0.491 | 0.927 | 0.743 | 0.414 | 0.423 |
| | 336 | 0.421 | 0.432 | 0.442 | 0.433 | 0.404 | 0.431 | 0.430 | 0.427 | 0.424 | 0.429 | 0.431 | 0.442 | 0.458 | 0.458 | 0.481 | 0.486 | 0.973 | 0.785 | 0.434 | 0.440 |
| | 720 | 0.445 | 0.470 | 0.477 | 0.456 | 0.495 | 0.495 | 0.442 | 0.457 | 0.425 | 0.448 | 0.472 | 0.494 | 0.473 | 0.484 | 0.551 | 0.525 | 1.048 | 0.834 | 0.431 | 0.454 |
| | Avg | 0.409 | 0.430 | 0.427 | 0.426 | 0.422 | 0.440 | 0.408 | 0.423 | **0.407** | **0.422** | 0.418 | 0.438 | 0.430 | 0.444 | 0.492 | 0.490 | 0.913 | 0.748 | 0.415 | 0.430 |
| ETTh2 | 96 | 0.268 | 0.337 | 0.285 | 0.342 | 0.270 | 0.339 | 0.268 | 0.328 | 0.271 | 0.337 | 0.303 | 0.368 | 0.298 | 0.349 | 0.382 | 0.420 | 1.497 | 0.934 | 0.271 | 0.337 |
| | 192 | 0.321 | 0.376 | 0.354 | 0.389 | 0.317 | 0.374 | 0.329 | 0.375 | 0.330 | 0.374 | 0.393 | 0.425 | 0.371 | 0.399 | 0.407 | 0.434 | 4.498 | 1.699 | 0.331 | 0.375 |
| | 336 | 0.331 | 0.394 | 0.373 | 0.407 | 0.320 | 0.382 | 0.368 | 0.409 | 0.353 | 0.395 | 0.510 | 0.497 | 0.418 | 0.433 | 0.392 | 0.438 | 4.385 | 1.743 | 0.354 | 0.396 |
| | 720 | 0.348 | 0.402 | 0.406 | 0.441 | 0.405 | 0.443 | 0.372 | 0.420 | 0.378 | 0.421 | 0.808 | 0.637 | 0.465 | 0.474 | 0.451 | 0.469 | 4.579 | 1.838 | 0.377 | 0.423 |
| | Avg | **0.317** | **0.377** | 0.354 | 0.394 | 0.328 | 0.385 | 0.334 | 0.383 | 0.333 | 0.382 | 0.504 | 0.482 | 0.388 | 0.414 | 0.408 | 0.440 | 3.740 | 1.554 | 0.333 | 0.383 |
| ETTm1 | 96 | 0.291 | 0.349 | 0.292 | 0.346 | 0.297 | 0.354 | 0.272 | 0.334 | 0.306 | 0.349 | 0.304 | 0.347 | 0.302 | 0.353 | 0.331 | 0.375 | 0.567 | 0.502 | 0.309 | 0.352 |
| | 192 | 0.334 | 0.374 | 0.332 | 0.372 | 0.329 | 0.373 | 0.310 | 0.358 | 0.338 | 0.367 | 0.335 | 0.366 | 0.335 | 0.373 | 0.394 | 0.409 | 0.626 | 0.546 | 0.338 | 0.368 |
| | 336 | 0.367 | 0.398 | 0.366 | 0.394 | 0.353 | 0.390 | 0.352 | 0.384 | 0.368 | 0.384 | 0.366 | 0.385 | 0.372 | 0.399 | 0.414 | 0.431 | 0.752 | 0.641 | 0.366 | 0.385 |
| | 720 | 0.425 | 0.418 | 0.417 | 0.421 | 0.411 | 0.416 | 0.383 | 0.411 | 0.421 | 0.413 | 0.420 | 0.417 | 0.442 | 0.437 | 0.454 | 0.457 | 0.952 | 0.745 | 0.415 | 0.412 |
| | Avg | 0.354 | 0.385 | 0.352 | 0.383 | 0.348 | 0.383 | **0.329** | **0.372** | 0.358 | 0.378 | 0.356 | 0.379 | 0.363 | 0.391 | 0.398 | 0.418 | 0.724 | 0.609 | 0.357 | 0.379 |
| ETTm2 | 96 | 0.164 | 0.254 | 0.173 | 0.262 | 0.169 | 0.263 | 0.161 | 0.253 | 0.165 | 0.254 | 0.169 | 0.265 | 0.169 | 0.260 | 0.191 | 0.277 | 0.369 | 0.464 | 0.163 | 0.253 |
| | 192 | 0.228 | 0.301 | 0.229 | 0.301 | 0.231 | 0.306 | 0.219 | 0.293 | 0.219 | 0.291 | 0.226 | 0.306 | 0.243 | 0.308 | 0.251 | 0.321 | 0.571 | 0.573 | 0.222 | 0.298 |
| | 336 | 0.279 | 0.335 | 0.286 | 0.341 | 0.286 | 0.339 | 0.271 | 0.329 | 0.272 | 0.326 | 0.298 | 0.362 | 0.297 | 0.353 | 0.319 | 0.364 | 1.193 | 0.813 | 0.268 | 0.326 |
| | 720 | 0.362 | 0.388 | 0.378 | 0.401 | 0.364 | 0.391 | 0.353 | 0.379 | 0.359 | 0.381 | 0.408 | 0.434 | 0.381 | 0.395 | 0.388 | 0.408 | 3.289 | 1.339 | 0.349 | 0.378 |
| | Avg | 0.258 | 0.319 | 0.266 | 0.326 | 0.263 | 0.325 | **0.251** | **0.313** | 0.254 | **0.313** | 0.275 | 0.342 | 0.273 | 0.329 | 0.287 | 0.343 | 1.356 | 0.797 | **0.251** | 0.314 |
| Electricity | 96 | 0.126 | 0.223 | 0.139 | 0.238 | 0.131 | 0.226 | 0.131 | 0.224 | 0.140 | 0.238 | 0.141 | 0.241 | 0.131 | 0.228 | 0.181 | 0.286 | 0.281 | 0.375 | 0.137 | 0.235 |
| | 192 | 0.144 | 0.240 | 0.153 | 0.251 | 0.145 | 0.240 | 0.152 | 0.241 | 0.153 | 0.250 | 0.154 | 0.254 | 0.149 | 0.246 | 0.191 | 0.294 | 0.295 | 0.389 | 0.151 | 0.248 |
| | 336 | 0.157 | 0.255 | 0.169 | 0.266 | 0.162 | 0.139 | 0.160 | 0.248 | 0.169 | 0.266 | 0.169 | 0.271 | 0.166 | 0.262 | 0.203 | 0.304 | 0.312 | 0.405 | 0.167 | 0.264 |
| | 720 | 0.181 | 0.279 | 0.206 | 0.297 | 0.199 | 0.290 | 0.192 | 0.298 | 0.208 | 0.298 | 0.204 | 0.304 | 0.203 | 0.293 | 0.294 | 0.371 | 0.309 | 0.396 | 0.206 | 0.296 |
| | Avg | **0.152** | 0.249 | 0.167 | 0.263 | 0.159 | **0.224** | 0.158 | 0.252 | 0.168 | 0.263 | 0.167 | 0.268 | 0.162 | 0.257 | 0.217 | 0.314 | 0.299 | 0.391 | 0.165 | 0.261 |
| Traffic | 96 | 0.361 | 0.255 | 0.388 | 0.282 | 0.371 | 0.258 | 0.362 | 0.248 | 0.398 | 0.277 | 0.411 | 0.294 | 0.364 | 0.256 | 0.601 | 0.320 | 0.708 | 0.406 | 0.387 | 0.272 |
| | 192 | 0.380 | 0.263 | 0.407 | 0.290 | 0.388 | 0.268 | 0.374 | 0.247 | 0.409 | 0.280 | 0.421 | 0.298 | 0.382 | 0.263 | 0.609 | 0.328 | 0.699 | 0.401 | 0.398 | 0.274 |
| | 336 | 0.393 | 0.269 | 0.412 | 0.294 | 0.397 | 0.272 | 0.385 | 0.271 | 0.418 | 0.285 | 0.431 | 0.304 | 0.392 | 0.270 | 0.619 | 0.330 | 0.699 | 0.398 | 0.411 | 0.281 |
| | 720 | 0.434 | 0.293 | 0.450 | 0.312 | 0.431 | 0.288 | 0.430 | 0.288 | 0.456 | 0.306 | 0.467 | 0.324 | 0.430 | 0.290 | 0.657 | 0.348 | 0.714 | 0.398 | 0.449 | 0.301 |
| | Avg | 0.392 | 0.270 | 0.414 | 0.294 | 0.397 | 0.272 | **0.388** | **0.264** | 0.420 | 0.287 | 0.433 | 0.305 | 0.392 | 0.270 | 0.622 | 0.332 | 0.705 | 0.401 | 0.411 | 0.282 |
| Exchange | 96 | 0.081 | 0.197 | 0.089 | 0.207 | 0.082 | 0.201 | 0.088 | 0.208 | 0.100 | 0.225 | 0.110 | 0.252 | 0.101 | 0.224 | 0.206 | 0.342 | 0.654 | 0.659 | 0.095 | 0.221 |
| | 192 | 0.171 | 0.292 | 0.185 | 0.305 | 0.171 | 0.293 | 0.182 | 0.303 | 0.201 | 0.326 | 0.238 | 0.373 | 0.215 | 0.335 | 0.391 | 0.472 | 0.741 | 0.691 | 0.200 | 0.323 |
| | 336 | 0.311 | 0.402 | 0.335 | 0.421 | 0.319 | 0.407 | 0.334 | 0.419 | 0.350 | 0.437 | 0.466 | 0.525 | 0.296 | 0.399 | 0.606 | 0.595 | 1.042 | 0.835 | 0.369 | 0.444 |
| | 720 | 0.822 | 0.682 | 0.884 | 0.705 | 0.888 | 0.739 | 0.879 | 0.704 | 0.920 | 0.728 | 1.166 | 0.821 | 1.058 | 0.775 | 1.601 | 0.961 | 2.190 | 1.189 | 0.994 | 0.750 |
| | Avg | **0.346** | **0.393** | 0.373 | 0.410 | 0.365 | 0.410 | 0.371 | 0.409 | 0.393 | 0.429 | 0.495 | 0.493 | 0.418 | 0.433 | 0.701 | 0.593 | 1.157 | 0.844 | 0.415 | 0.435 |
| Average | | **0.308** | 0.337 | 0.324 | 0.346 | 0.320 | 0.341 | **0.308** | **0.334** | 0.322 | 0.344 | 0.361 | 0.375 | 0.331 | 0.350 | 0.422 | 0.402 | 1.147 | 0.711 | 0.323 | 0.345 |

Table 14: Full results of 96 look-back window length in long-term forecasting task. (* means former, Station means the Non-stationary Transformer.) The standard deviation is within 0.5%. We copied the results of iTransformer from TSLANet [27], Pyraformer from TimesNet [13], reproduced FITS by https://github.com/VEWOXIC/FITS, and copied the others from GPT4TS [20]. **Red**: best, Blue: second best.

| Look-back | | \multicolumn{28}{c}{96} | | | | | | | | | | | | | | | | | | | | | | | | | | |
|---|---|---|---|---|---|---|---|---|---|---|---|---|---|---|---|---|---|---|---|---|---|---|---|---|---|---|---|---|---|
| Methods | | Peri-mid Former | | iTrans* [26] | | FITS [21] | | Dlinear [14] | | PatchTST [19] | | TimesNet [13] | | Pyra* [10] | | FED* [17] | | Auto* [12] | | Station [11] | | ETS* [29] | | LightTS [15] | | In* [16] | | Re* [28] | |
| Metrics | | MSE | MAE | MSE | MAE | MSE | MAE | MSE | MAE | MSE | MAE | MSE | MAE | MSE | MAE | MSE | MAE | MSE | MAE | MSE | MAE | MSE | MAE | MSE | MAE | MSE | MAE | MSE | MAE |
| Weather | 96 | 0.155 | 0.200 | 0.174 | 0.214 | 0.197 | 0.237 | 0.176 | 0.237 | 0.149 | 0.198 | 0.172 | 0.220 | 0.622 | 0.556 | 0.217 | 0.296 | 0.266 | 0.336 | 0.173 | 0.223 | 0.197 | 0.281 | 0.182 | 0.242 | 0.300 | 0.384 | 0.689 | 0.596 |
| | 192 | 0.203 | 0.244 | 0.221 | 0.254 | 0.241 | 0.272 | 0.220 | 0.282 | 0.194 | 0.241 | 0.219 | 0.261 | 0.739 | 0.624 | 0.276 | 0.336 | 0.307 | 0.367 | 0.245 | 0.285 | 0.237 | 0.312 | 0.227 | 0.287 | 0.598 | 0.544 | 0.752 | 0.638 |
| | 336 | 0.262 | 0.289 | 0.278 | 0.296 | 0.293 | 0.308 | 0.265 | 0.319 | 0.245 | 0.282 | 0.280 | 0.306 | 1.004 | 0.753 | 0.339 | 0.380 | 0.359 | 0.395 | 0.321 | 0.338 | 0.298 | 0.353 | 0.282 | 0.334 | 0.578 | 0.523 | 0.639 | 0.596 |
| | 720 | 0.345 | 0.344 | 0.358 | 0.349 | 0.365 | 0.354 | 0.333 | 0.362 | 0.314 | 0.334 | 0.365 | 0.359 | 1.420 | 0.934 | 0.403 | 0.428 | 0.419 | 0.428 | 0.414 | 0.410 | 0.352 | 0.288 | 0.352 | 0.386 | 1.059 | 0.741 | 1.130 | 0.792 |
| | Avg | 0.241 | 0.269 | 0.258 | 0.278 | 0.274 | 0.293 | 0.248 | 0.300 | **0.225** | **0.264** | 0.259 | 0.287 | 0.946 | 0.717 | 0.309 | 0.360 | 0.338 | 0.382 | 0.288 | 0.314 | 0.271 | 0.334 | 0.261 | 0.312 | 0.634 | 0.548 | 0.803 | 0.656 |
| ETTh1 | 96 | 0.373 | 0.396 | 0.386 | 0.405 | 0.385 | 0.393 | 0.375 | 0.399 | 0.370 | 0.399 | 0.384 | 0.402 | 0.664 | 0.612 | 0.376 | 0.419 | 0.449 | 0.459 | 0.513 | 0.491 | 0.494 | 0.479 | 0.424 | 0.432 | 0.865 | 0.713 | 0.837 | 0.728 |
| | 192 | 0.425 | 0.429 | 0.441 | 0.436 | 0.435 | 0.422 | 0.405 | 0.416 | 0.413 | 0.421 | 0.436 | 0.429 | 0.790 | 0.681 | 0.420 | 0.448 | 0.500 | 0.482 | 0.534 | 0.504 | 0.538 | 0.504 | 0.475 | 0.462 | 1.008 | 0.792 | 0.923 | 0.766 |
| | 336 | 0.464 | 0.445 | 0.487 | 0.458 | 0.475 | 0.444 | 0.439 | 0.443 | 0.422 | 0.436 | 0.491 | 0.469 | 0.891 | 0.738 | 0.459 | 0.465 | 0.521 | 0.496 | 0.588 | 0.535 | 0.574 | 0.521 | 0.518 | 0.488 | 1.107 | 0.809 | 1.097 | 0.835 |
| | 720 | 0.479 | 0.467 | 0.503 | 0.491 | 0.463 | 0.459 | 0.472 | 0.490 | 0.447 | 0.466 | 0.521 | 0.500 | 0.963 | 0.782 | 0.506 | 0.507 | 0.514 | 0.512 | 0.643 | 0.616 | 0.562 | 0.535 | 0.547 | 0.533 | 1.181 | 0.865 | 1.257 | 0.889 |
| | Avg | 0.435 | 0.434 | 0.454 | 0.448 | 0.440 | **0.429** | 0.422 | 0.437 | **0.413** | 0.430 | 0.458 | 0.450 | 0.827 | 0.703 | 0.440 | 0.460 | 0.496 | 0.487 | 0.570 | 0.537 | 0.542 | 0.510 | 0.491 | 0.479 | 1.040 | 0.795 | 1.029 | 0.805 |
| ETTh2 | 96 | 0.287 | 0.337 | 0.297 | 0.349 | 0.292 | 0.339 | 0.289 | 0.353 | 0.274 | 0.336 | 0.340 | 0.374 | 0.645 | 0.597 | 0.358 | 0.397 | 0.346 | 0.388 | 0.476 | 0.458 | 0.340 | 0.391 | 0.397 | 0.437 | 3.755 | 1.525 | 2.626 | 1.317 |
| | 192 | 0.368 | 0.387 | 0.380 | 0.400 | 0.377 | 0.391 | 0.383 | 0.418 | 0.339 | 0.379 | 0.402 | 0.414 | 0.788 | 0.683 | 0.429 | 0.439 | 0.456 | 0.452 | 0.512 | 0.493 | 0.430 | 0.439 | 0.520 | 0.504 | 5.602 | 1.931 | 11.120 | 2.979 |
| | 336 | 0.414 | 0.424 | 0.428 | 0.432 | 0.416 | 0.425 | 0.448 | 0.465 | 0.329 | 0.380 | 0.452 | 0.452 | 0.907 | 0.747 | 0.496 | 0.487 | 0.482 | 0.486 | 0.552 | 0.551 | 0.485 | 0.479 | 0.626 | 5.559 | 4.721 | 1.833 | 3.233 | 2.769 |
| | 720 | 0.397 | 0.425 | 0.427 | 0.445 | 0.417 | 0.436 | 0.605 | 0.551 | 0.379 | 0.422 | 0.462 | 0.468 | 0.963 | 0.783 | 0.463 | 0.474 | 0.515 | 0.511 | 0.562 | 0.560 | 0.500 | 0.497 | 0.863 | 0.672 | 3.647 | 1.625 | 3.874 | 1.697 |
| | Avg | 0.367 | 0.393 | 0.383 | 0.407 | 0.376 | 0.398 | 0.431 | 0.446 | **0.330** | **0.379** | 0.415 | 0.427 | 0.826 | 0.703 | 0.437 | 0.449 | 0.450 | 0.459 | 0.526 | 0.516 | 0.439 | 0.452 | 0.602 | 0.543 | 4.431 | 7.000 | 1.729 | 6.736 |
| ETTm1 | 96 | 0.330 | 0.368 | 0.334 | 0.368 | 0.353 | 0.375 | 0.299 | 0.343 | 0.290 | 0.342 | 0.338 | 0.375 | 0.543 | 0.510 | 0.379 | 0.419 | 0.505 | 0.475 | 0.386 | 0.398 | 0.375 | 0.398 | 0.370 | 0.400 | 0.677 | 0.571 | 0.533 | 0.528 |
| | 192 | 0.371 | 0.388 | 0.377 | 0.391 | 0.392 | 0.393 | 0.335 | 0.365 | 0.332 | 0.369 | 0.374 | 0.387 | 0.556 | 0.537 | 0.426 | 0.441 | 0.553 | 0.496 | 0.459 | 0.444 | 0.408 | 0.410 | 0.400 | 0.407 | 0.795 | 0.669 | 0.658 | 0.592 |
| | 336 | 0.402 | 0.409 | 0.426 | 0.420 | 0.425 | 0.414 | 0.369 | 0.386 | 0.366 | 0.392 | 0.410 | 0.411 | 0.754 | 0.655 | 0.445 | 0.459 | 0.621 | 0.537 | 0.495 | 0.464 | 0.435 | 0.428 | 0.438 | 0.438 | 1.212 | 0.871 | 0.898 | 0.721 |
| | 720 | 0.466 | 0.445 | 0.491 | 0.459 | 0.486 | 0.448 | 0.425 | 0.421 | 0.416 | 0.420 | 0.478 | 0.450 | 0.908 | 0.724 | 0.543 | 0.490 | 0.671 | 0.561 | 0.585 | 0.516 | 0.499 | 0.462 | 0.527 | 0.502 | 1.166 | 0.823 | 1.102 | 0.841 |
| | Avg | 0.392 | 0.402 | 0.407 | 0.410 | 0.414 | 0.408 | 0.357 | **0.378** | **0.351** | 0.380 | 0.400 | 0.406 | 0.691 | 0.607 | 0.448 | 0.452 | 0.588 | 0.517 | 0.481 | 0.456 | 0.429 | 0.425 | 0.435 | 0.437 | 0.961 | 0.734 | 0.799 | 0.671 |
| ETTm2 | 96 | 0.171 | 0.254 | 0.180 | 0.264 | 0.183 | 0.266 | 0.167 | 0.269 | 0.165 | 0.255 | 0.187 | 0.267 | 0.435 | 0.507 | 0.203 | 0.287 | 0.255 | 0.339 | 0.192 | 0.274 | 0.189 | 0.280 | 0.209 | 0.308 | 0.365 | 0.453 | 0.658 | 0.619 |
| | 192 | 0.240 | 0.299 | 0.250 | 0.309 | 0.247 | 0.305 | 0.224 | 0.303 | 0.220 | 0.292 | 0.249 | 0.309 | 0.730 | 0.673 | 0.269 | 0.328 | 0.281 | 0.340 | 0.280 | 0.339 | 10.253 | 0.319 | 0.311 | 0.382 | 0.533 | 0.563 | 1.078 | 0.827 |
| | 336 | 0.306 | 0.341 | 0.311 | 0.348 | 0.307 | 0.342 | 0.281 | 0.342 | 0.274 | 0.329 | 0.321 | 0.351 | 1.201 | 0.845 | 0.325 | 0.366 | 0.339 | 0.372 | 0.334 | 0.361 | 0.314 | 0.357 | 0.442 | 0.466 | 1.363 | 0.887 | 1.549 | 0.972 |
| | 720 | 0.404 | 0.397 | 0.412 | 0.407 | 0.407 | 0.397 | 0.397 | 0.421 | 0.362 | 0.385 | 0.408 | 0.403 | 3.625 | 1.451 | 0.421 | 0.415 | 0.433 | 0.432 | 0.417 | 0.413 | 0.414 | 0.413 | 0.675 | 0.587 | 3.379 | 1.338 | 2.631 | 1.242 |
| | Avg | 0.280 | **0.323** | 0.288 | 0.332 | 0.286 | 0.328 | 0.267 | 0.333 | **0.255** | 0.315 | 0.291 | 0.333 | 1.498 | 0.869 | 0.305 | 0.349 | 0.327 | 0.371 | 0.306 | 0.347 | 0.293 | 0.342 | 0.409 | 0.436 | 1.410 | 0.810 | 1.479 | 0.915 |
| Electricity | 96 | 0.142 | 0.236 | 0.148 | 0.240 | 0.197 | 0.274 | 0.140 | 0.237 | 0.129 | 0.222 | 0.168 | 0.272 | 0.386 | 0.449 | 0.193 | 0.308 | 0.201 | 0.317 | 0.169 | 0.273 | 0.187 | 0.304 | 0.207 | 0.307 | 0.274 | 0.368 | 0.312 | 0.402 |
| | 192 | 0.159 | 0.251 | 0.162 | 0.253 | 0.197 | 0.276 | 0.153 | 0.249 | 0.157 | 0.240 | 0.184 | 0.289 | 0.378 | 0.443 | 0.201 | 0.315 | 0.222 | 0.334 | 0.182 | 0.286 | 0.199 | 0.315 | 0.213 | 0.316 | 0.296 | 0.386 | 0.348 | 0.430 |
| | 336 | 0.176 | 0.270 | 0.178 | 0.269 | 0.212 | 0.293 | 0.160 | 0.267 | 0.163 | 0.259 | 0.198 | 0.300 | 0.376 | 0.443 | 0.214 | 0.329 | 0.230 | 0.333 | 0.200 | 0.304 | 0.212 | 0.329 | 0.230 | 0.333 | 0.300 | 0.394 | 0.350 | 0.433 |
| | 720 | 0.204 | 0.307 | 0.225 | 0.317 | 0.253 | 0.325 | 0.203 | 0.301 | 0.197 | 0.290 | 0.220 | 0.320 | 0.376 | 0.445 | 0.246 | 0.355 | 0.254 | 0.361 | 0.222 | 0.321 | 0.233 | 0.345 | 0.265 | 0.360 | 0.373 | 0.439 | 0.340 | 0.42 |
| | Avg | 0.170 | 0.266 | 0.178 | 0.270 | 0.215 | 0.292 | 0.166 | 0.263 | **0.161** | **0.252** | 0.192 | 0.295 | 0.379 | 0.445 | 0.214 | 0.327 | 0.227 | 0.334 | 0.193 | 0.296 | 0.208 | 0.323 | 0.229 | 0.329 | 0.311 | 0.397 | 0.338 | 0.422 |
| Traffic | 96 | 0.453 | 0.300 | 0.395 | 0.268 | 0.642 | 0.388 | 0.410 | 0.282 | 0.360 | 0.249 | 0.593 | 0.321 | 0.867 | 0.468 | 0.587 | 0.366 | 0.613 | 0.388 | 0.612 | 0.338 | 0.607 | 0.392 | 0.615 | 0.391 | 0.719 | 0.391 | 0.732 | 0.423 |
| | 192 | 0.459 | 0.301 | 0.417 | 0.276 | 0.597 | 0.362 | 0.423 | 0.287 | 0.379 | 0.256 | 0.617 | 0.336 | 0.869 | 0.467 | 0.604 | 0.373 | 0.616 | 0.382 | 0.613 | 0.340 | 0.621 | 0.399 | 0.601 | 0.382 | 0.696 | 0.379 | 0.733 | 0.420 |
| | 336 | 0.477 | 0.310 | 0.433 | 0.283 | 0.603 | 0.365 | 0.436 | 0.296 | 0.392 | 0.264 | 0.629 | 0.336 | 0.881 | 0.469 | 0.621 | 0.383 | 0.622 | 0.337 | 0.618 | 0.328 | 0.622 | 0.396 | 0.613 | 0.386 | 0.777 | 0.420 | 0.742 | 0.420 |
| | 720 | 0.510 | 0.329 | 0.467 | 0.302 | 0.641 | 0.384 | 0.466 | 0.315 | 0.432 | 0.286 | 0.640 | 0.350 | 0.896 | 0.473 | 0.626 | 0.382 | 0.660 | 0.408 | 0.653 | 0.355 | 0.632 | 0.396 | 0.658 | 0.407 | 0.864 | 0.472 | 0.755 | 0.423 |
| | Avg | 0.475 | 0.310 | 0.428 | 0.282 | 0.621 | 0.375 | 0.433 | 0.295 | **0.390** | **0.263** | 0.620 | 0.336 | 0.878 | 0.469 | 0.610 | 0.376 | 0.628 | 0.379 | 0.624 | 0.340 | 0.621 | 0.396 | 0.623 | 0.392 | 0.764 | 0.416 | 0.741 | 0.422 |
| Exchange | 96 | 0.082 | 0.198 | 0.086 | 0.206 | 0.087 | 0.208 | 0.088 | 0.218 | 0.088 | 0.205 | 0.107 | 0.234 | 1.748 | 1.105 | 0.148 | 0.278 | 0.197 | 0.323 | 0.111 | 0.237 | 0.085 | 0.204 | 0.116 | 0.262 | 0.847 | 0.752 | 1.065 | 0.829 |
| | 192 | 0.172 | 0.295 | 0.177 | 0.299 | 0.178 | 0.302 | 0.176 | 0.315 | 0.175 | 0.298 | 0.226 | 0.344 | 1.874 | 1.151 | 0.271 | 0.380 | 0.300 | 0.369 | 0.219 | 0.335 | 0.182 | 0.303 | 0.215 | 0.359 | 1.204 | 0.895 | 1.188 | 0.906 |
| | 336 | 0.319 | 0.407 | 0.331 | 0.417 | 0.325 | 0.413 | 0.313 | 0.427 | 0.302 | 0.398 | 0.367 | 0.448 | 1.943 | 1.172 | 0.460 | 0.500 | 0.509 | 0.524 | 0.421 | 0.476 | 0.348 | 0.428 | 0.377 | 0.466 | 1.672 | 1.036 | 1.357 | 0.976 |
| | 720 | 0.846 | 0.692 | 0.847 | 0.691 | 0.843 | 0.694 | 0.839 | 0.695 | 0.871 | 0.703 | 0.964 | 0.746 | 2.085 | 1.206 | 1.195 | 0.841 | 1.447 | 0.941 | 1.092 | 0.769 | 1.025 | 0.774 | 0.831 | 0.699 | 2.478 | 1.310 | 1.510 | 1.016 |
| | Avg | 0.355 | **0.398** | 0.360 | 0.403 | 0.358 | 0.404 | **0.354** | 0.414 | 0.359 | 0.401 | 0.416 | 0.443 | 1.913 | 1.159 | 0.519 | 0.500 | 0.613 | 0.539 | 0.461 | 0.454 | 0.410 | 0.427 | 0.385 | 0.447 | 1.550 | 0.998 | 1.280 | 0.932 |
| Average | | 0.339 | 0.349 | 0.345 | 0.353 | 0.373 | 0.366 | 0.335 | 0.358 | **0.311** | **0.335** | 0.381 | 0.372 | 0.995 | 0.709 | 0.410 | 0.409 | 0.458 | 0.433 | 0.431 | 0.408 | 0.402 | 0.401 | 0.429 | 0.422 | 0.953 | 0.803 | 1.650 | 0.877 |

Table 15: Full results for the imputation task. We randomly mask 12.5%, 25%, 37.5% and 50% time points to compare the model performance under different missing degrees. (∗ means former, Station means the Non-stationary Transformer.) The standard deviation is within 0.5%. We reproduced the results of Pyraformer by https://github.com/thuml/Time-Series-Library, and copied the others from GPT4TS [20]. **Red**: best, Blue: second best.

| Models | | Peri-midFormer (ours) | | GPT4TS [20] | | Timesnet [13] | | PatchTST [19] | | ETS∗ [29] | | LightTS [15] | | DLinear [14] | | FED∗ [17] | | Station [11] | | Auto∗ [12] | | Pyra∗ [10] | | In∗ [16] | | Re∗ [28] | |
|---|---|---|---|---|---|---|---|---|---|---|---|---|---|---|---|---|---|---|---|---|---|---|---|---|---|---|---|
| Mask Ratio | | MSE | MAE | MSE | MAE | MSE | MAE | MSE | MAE | MSE | MAE | MSE | MAE | MSE | MAE | MSE | MAE | MSE | MAE | MSE | MAE | MSE | MAE | MSE | MAE | MSE | MAE |
| ETTm1 12.5% | | 0.030 | 0.108 | 0.017 | 0.085 | 0.019 | 0.092 | 0.041 | 0.130 | 0.067 | 0.188 | 0.075 | 0.180 | 0.058 | 0.162 | 0.035 | 0.135 | 0.026 | 0.107 | 0.034 | 0.124 | 0.670 | 0.541 | 0.047 | 0.155 | 0.032 | 0.126 |
| ETTm1 25% | | 0.031 | 0.112 | 0.022 | 0.096 | 0.023 | 0.101 | 0.044 | 0.135 | 0.096 | 0.229 | 0.093 | 0.206 | 0.080 | 0.193 | 0.052 | 0.166 | 0.032 | 0.119 | 0.046 | 0.144 | 0.689 | 0.553 | 0.063 | 0.180 | 0.042 | 0.146 |
| ETTm1 37.5% | | 0.034 | 0.117 | 0.029 | 0.111 | 0.029 | 0.111 | 0.049 | 0.143 | 0.133 | 0.271 | 0.113 | 0.231 | 0.103 | 0.219 | 0.069 | 0.191 | 0.039 | 0.131 | 0.057 | 0.161 | 0.737 | 0.581 | 0.079 | 0.200 | 0.063 | 0.182 |
| ETTm1 50% | | 0.041 | 0.125 | 0.040 | 0.128 | 0.036 | 0.124 | 0.055 | 0.151 | 0.186 | 0.323 | 0.134 | 0.255 | 0.132 | 0.248 | 0.089 | 0.218 | 0.047 | 0.145 | 0.067 | 0.174 | 0.770 | 0.605 | 0.093 | 0.218 | 0.082 | 0.208 |
| ETTm1 Avg | | 0.034 | 0.116 | 0.028 | **0.105** | **0.027** | 0.107 | 0.047 | 0.140 | 0.120 | 0.253 | 0.104 | 0.218 | 0.093 | 0.206 | 0.062 | 0.177 | 0.036 | 0.126 | 0.051 | 0.150 | 0.717 | 0.570 | 0.071 | 0.188 | 0.055 | 0.166 |
| ETTm2 12.5% | | 0.022 | 0.081 | 0.017 | 0.076 | 0.018 | 0.080 | 0.026 | 0.094 | 0.108 | 0.239 | 0.034 | 0.127 | 0.062 | 0.166 | 0.056 | 0.159 | 0.021 | 0.088 | 0.023 | 0.092 | 0.394 | 0.470 | 0.133 | 0.270 | 0.108 | 0.228 |
| ETTm2 25% | | 0.024 | 0.084 | 0.020 | 0.080 | 0.020 | 0.085 | 0.028 | 0.099 | 0.164 | 0.294 | 0.042 | 0.143 | 0.085 | 0.196 | 0.080 | 0.195 | 0.024 | 0.096 | 0.026 | 0.101 | 0.421 | 0.482 | 0.135 | 0.272 | 0.136 | 0.262 |
| ETTm2 37.5% | | 0.026 | 0.089 | 0.022 | 0.087 | 0.023 | 0.091 | 0.030 | 0.104 | 0.237 | 0.356 | 0.051 | 0.159 | 0.106 | 0.222 | 0.110 | 0.231 | 0.027 | 0.103 | 0.030 | 0.108 | 0.478 | 0.521 | 0.155 | 0.293 | 0.175 | 0.300 |
| ETTm2 50% | | 0.029 | 0.095 | 0.025 | 0.095 | 0.026 | 0.098 | 0.034 | 0.110 | 0.323 | 0.421 | 0.059 | 0.174 | 0.131 | 0.247 | 0.156 | 0.276 | 0.030 | 0.108 | 0.035 | 0.119 | 0.568 | 0.560 | 0.200 | 0.333 | 0.211 | 0.329 |
| ETTm2 Avg | | 0.025 | 0.087 | **0.021** | **0.084** | 0.022 | 0.088 | 0.029 | 0.102 | 0.208 | 0.327 | 0.046 | 0.151 | 0.096 | 0.208 | 0.101 | 0.215 | 0.026 | 0.099 | 0.029 | 0.105 | 0.465 | 0.508 | 0.156 | 0.292 | 0.157 | 0.280 |
| ETTh1 12.5% | | 0.060 | 0.164 | 0.043 | 0.140 | 0.057 | 0.159 | 0.093 | 0.201 | 0.126 | 0.263 | 0.240 | 0.345 | 0.151 | 0.267 | 0.070 | 0.190 | 0.060 | 0.165 | 0.074 | 0.182 | 0.857 | 0.609 | 0.114 | 0.234 | 0.074 | 0.194 |
| ETTh1 25% | | 0.069 | 0.181 | 0.054 | 0.156 | 0.069 | 0.178 | 0.107 | 0.217 | 0.169 | 0.304 | 0.265 | 0.364 | 0.180 | 0.292 | 0.106 | 0.236 | 0.080 | 0.189 | 0.090 | 0.203 | 0.829 | 0.672 | 0.140 | 0.262 | 0.102 | 0.227 |
| ETTh1 37.5% | | 0.092 | 0.200 | 0.072 | 0.180 | 0.084 | 0.196 | 0.120 | 0.230 | 0.220 | 0.347 | 0.296 | 0.382 | 0.215 | 0.318 | 0.124 | 0.258 | 0.102 | 0.212 | 0.109 | 0.222 | 0.830 | 0.675 | 0.174 | 0.293 | 0.135 | 0.261 |
| ETTh1 50% | | 0.112 | 0.220 | 0.107 | 0.216 | 0.102 | 0.215 | 0.141 | 0.248 | 0.293 | 0.402 | 0.334 | 0.404 | 0.257 | 0.347 | 0.165 | 0.299 | 0.133 | 0.240 | 0.137 | 0.248 | 0.854 | 0.691 | 0.215 | 0.325 | 0.179 | 0.298 |
| ETTh1 Avg | | 0.083 | 0.191 | **0.069** | **0.173** | 0.078 | 0.187 | 0.115 | 0.224 | 0.202 | 0.329 | 0.284 | 0.373 | 0.201 | 0.306 | 0.117 | 0.246 | 0.094 | 0.201 | 0.103 | 0.214 | 0.842 | 0.682 | 0.161 | 0.279 | 0.122 | 0.245 |
| ETTh2 12.5% | | 0.049 | 0.140 | 0.039 | 0.125 | 0.040 | 0.130 | 0.057 | 0.152 | 0.187 | 0.319 | 0.101 | 0.231 | 0.100 | 0.216 | 0.095 | 0.212 | 0.042 | 0.133 | 0.044 | 0.138 | 0.976 | 0.754 | 0.305 | 0.431 | 0.163 | 0.289 |
| ETTh2 25% | | 0.051 | 0.141 | 0.044 | 0.135 | 0.046 | 0.141 | 0.061 | 0.158 | 0.279 | 0.390 | 0.115 | 0.246 | 0.127 | 0.247 | 0.137 | 0.258 | 0.049 | 0.147 | 0.050 | 0.149 | 1.037 | 0.774 | 0.322 | 0.444 | 0.206 | 0.331 |
| ETTh2 37.5% | | 0.055 | 0.147 | 0.051 | 0.147 | 0.052 | 0.151 | 0.067 | 0.166 | 0.400 | 0.465 | 0.126 | 0.257 | 0.158 | 0.276 | 0.187 | 0.304 | 0.056 | 0.158 | 0.060 | 0.163 | 1.107 | 0.800 | 0.353 | 0.462 | 0.252 | 0.370 |
| ETTh2 50% | | 0.062 | 0.156 | 0.059 | 0.158 | 0.060 | 0.162 | 0.073 | 0.174 | 0.602 | 0.572 | 0.136 | 0.268 | 0.183 | 0.299 | 0.232 | 0.341 | 0.065 | 0.170 | 0.068 | 0.173 | 1.193 | 0.838 | 0.369 | 0.472 | 0.316 | 0.419 |
| ETTh2 Avg | | 0.054 | 0.146 | **0.048** | **0.141** | 0.049 | 0.146 | 0.065 | 0.163 | 0.367 | 0.436 | 0.119 | 0.250 | 0.142 | 0.259 | 0.163 | 0.279 | 0.053 | 0.152 | 0.055 | 0.156 | 1.079 | 0.792 | 0.337 | 0.452 | 0.234 | 0.352 |
| Electricity 12.5% | | 0.044 | 0.135 | 0.080 | 0.194 | 0.085 | 0.202 | 0.055 | 0.160 | 0.196 | 0.321 | 0.102 | 0.229 | 0.092 | 0.214 | 0.107 | 0.237 | 0.093 | 0.210 | 0.089 | 0.210 | 0.297 | 0.383 | 0.218 | 0.326 | 0.190 | 0.308 |
| Electricity 25% | | 0.052 | 0.149 | 0.087 | 0.203 | 0.089 | 0.206 | 0.065 | 0.175 | 0.207 | 0.332 | 0.121 | 0.252 | 0.118 | 0.247 | 0.120 | 0.251 | 0.097 | 0.214 | 0.096 | 0.220 | 0.294 | 0.380 | 0.219 | 0.326 | 0.197 | 0.312 |
| Electricity 37.5% | | 0.063 | 0.166 | 0.094 | 0.211 | 0.094 | 0.213 | 0.076 | 0.189 | 0.219 | 0.344 | 0.141 | 0.273 | 0.144 | 0.276 | 0.136 | 0.266 | 0.102 | 0.220 | 0.104 | 0.229 | 0.296 | 0.381 | 0.222 | 0.328 | 0.203 | 0.315 |
| Electricity 50% | | 0.079 | 0.189 | 0.101 | 0.220 | 0.100 | 0.221 | 0.091 | 0.208 | 0.235 | 0.357 | 0.160 | 0.293 | 0.175 | 0.305 | 0.158 | 0.284 | 0.108 | 0.228 | 0.113 | 0.239 | 0.299 | 0.383 | 0.228 | 0.331 | 0.210 | 0.319 |
| Electricity Avg | | **0.060** | **0.160** | 0.090 | 0.207 | 0.092 | 0.210 | 0.072 | 0.183 | 0.214 | 0.339 | 0.131 | 0.262 | 0.132 | 0.260 | 0.130 | 0.259 | 0.100 | 0.218 | 0.101 | 0.225 | 0.297 | 0.382 | 0.222 | 0.328 | 0.200 | 0.313 |
| Weather 12.5% | | 0.025 | 0.037 | 0.026 | 0.049 | 0.025 | 0.045 | 0.029 | 0.049 | 0.057 | 0.141 | 0.047 | 0.101 | 0.039 | 0.084 | 0.041 | 0.107 | 0.027 | 0.051 | 0.026 | 0.047 | 0.140 | 0.220 | 0.037 | 0.093 | 0.031 | 0.076 |
| Weather 25% | | 0.026 | 0.037 | 0.025 | 0.052 | 0.029 | 0.052 | 0.031 | 0.053 | 0.065 | 0.155 | 0.052 | 0.111 | 0.048 | 0.103 | 0.064 | 0.163 | 0.029 | 0.056 | 0.030 | 0.054 | 0.147 | 0.229 | 0.042 | 0.100 | 0.035 | 0.082 |
| Weather 37.5% | | 0.028 | 0.040 | 0.033 | 0.060 | 0.031 | 0.057 | 0.035 | 0.058 | 0.081 | 0.180 | 0.058 | 0.121 | 0.057 | 0.117 | 0.107 | 0.229 | 0.033 | 0.062 | 0.032 | 0.060 | 0.156 | 0.240 | 0.049 | 0.111 | 0.040 | 0.091 |
| Weather 50% | | 0.031 | 0.042 | 0.037 | 0.065 | 0.034 | 0.062 | 0.083 | 0.063 | 0.102 | 0.207 | 0.065 | 0.133 | 0.066 | 0.134 | 0.183 | 0.312 | 0.037 | 0.068 | 0.037 | 0.067 | 0.164 | 0.249 | 0.053 | 0.114 | 0.046 | 0.099 |
| Weather Avg | | **0.028** | **0.039** | 0.031 | 0.056 | 0.030 | 0.054 | 0.060 | 0.144 | 0.076 | 0.171 | 0.055 | 0.117 | 0.052 | 0.110 | 0.099 | 0.203 | 0.032 | 0.059 | 0.031 | 0.057 | 0.152 | 0.235 | 0.045 | 0.104 | 0.038 | 0.087 |
| Average | | **0.047** | **0.123** | 0.048 | 0.128 | 0.050 | 0.132 | 0.053 | 0.159 | 0.164 | 0.309 | 0.123 | 0.229 | 0.119 | 0.225 | 0.112 | 0.23 | 0.057 | 0.143 | 0.062 | 0.151 | 0.592 | 0.528 | 0.165 | 0.274 | 0.134 | 0.241 |

Table 16: Full results for the anomaly detection task. The P, R and F1 represent the precision, recall and F1-score (%) respectively. F1-score is the harmonic mean of precision and recall. A higher value of P, R and F1 indicates a better performance. (Station means the Non-stationary Transformer.) The standard deviation is within 1%. We copied the results from GPT4TS [20]. **Red**: best, Blue: second best.

| Datasets | | SMD | | | MSL | | | SMAP | | | SWaT | | | PSM | | | Avg F1 |
|---|---|---|---|---|---|---|---|---|---|---|---|---|---|---|---|---|---|
| Metrics | | P | R | F1 | P | R | F1 | P | R | F1 | P | R | F1 | P | R | F1 | (%) |
| Transformer | [48] | 83.58 | 76.13 | 79.56 | 71.57 | 87.37 | 78.68 | 89.37 | 57.12 | 69.70 | 68.84 | 96.53 | 80.37 | 62.75 | 96.56 | 76.07 | 76.88 |
| LogTrans | [47] | 83.46 | 70.13 | 76.21 | 73.05 | 87.37 | 79.57 | 89.15 | 57.59 | 69.97 | 68.67 | 97.32 | 80.52 | 63.06 | 98.00 | 76.74 | 76.60 |
| Reformer | [28] | 82.58 | 69.24 | 75.32 | 85.51 | 83.31 | 84.40 | 90.91 | 57.44 | 70.40 | 72.50 | 96.53 | 82.80 | 59.93 | 95.38 | 73.61 | 77.31 |
| Informer | [16] | 86.60 | 77.23 | 81.65 | 81.77 | 86.48 | 84.06 | 90.11 | 57.13 | 69.92 | 70.29 | 96.75 | 81.43 | 64.27 | 96.33 | 77.10 | 78.83 |
| Anomaly | [32] | 88.91 | 82.23 | 85.49 | 79.61 | 87.37 | 83.31 | 91.85 | 58.11 | 71.18 | 72.51 | 97.32 | 83.10 | 68.35 | 94.72 | 79.40 | 80.50 |
| Pyraformer | [10] | 85.61 | 80.61 | 83.04 | 83.81 | 85.93 | 84.86 | 92.54 | 57.71 | 71.09 | 87.92 | 96.00 | 91.78 | 71.67 | 96.02 | 82.08 | 82.57 |
| Autoformer | [12] | 88.06 | 82.35 | 85.11 | 77.27 | 80.92 | 79.05 | 90.40 | 58.62 | 71.12 | 89.85 | 95.81 | 92.74 | 99.08 | 88.15 | 93.29 | 84.26 |
| Station | [11] | 88.33 | 81.21 | 84.62 | 68.55 | 89.14 | 77.50 | 89.37 | 59.02 | 71.09 | 68.03 | 96.75 | 79.88 | 97.82 | 96.76 | 97.29 | 82.08 |
| DLinear | [14] | 83.62 | 71.52 | 77.10 | 84.34 | 85.42 | 84.88 | 92.32 | 55.41 | 69.26 | 80.91 | 95.30 | 87.52 | 98.28 | 89.26 | 93.55 | 82.46 |
| LightTS | [41] | 87.10 | 78.42 | 82.53 | 82.40 | 75.78 | 78.95 | 92.58 | 55.27 | 69.21 | 91.98 | 94.72 | 93.33 | 98.37 | 95.97 | 97.15 | 84.23 |
| ETSformer | [29] | 87.44 | 79.23 | 83.13 | 85.13 | 84.93 | 85.03 | 92.25 | 55.75 | 69.50 | 90.02 | 80.36 | 84.91 | 99.31 | 85.28 | 91.76 | 82.87 |
| FEDformer | [17] | 87.95 | 82.39 | 85.08 | 77.14 | 80.07 | 78.57 | 90.47 | 58.10 | 70.76 | 90.17 | 96.42 | 93.19 | 97.31 | 97.16 | 97.23 | 84.97 |
| PatchTST | [19] | 87.26 | 82.14 | 84.62 | 88.34 | 70.96 | 78.70 | 90.64 | 55.46 | 68.82 | 91.1 | 80.94 | 85.72 | 98.84 | 93.47 | 96.08 | 82.79 |
| TimesNet | [13] | 87.91 | 81.54 | 84.61 | 89.54 | 75.36 | 81.84 | 90.14 | 56.4 | 69.39 | 90.75 | 95.4 | 93.02 | 98.51 | 96.20 | 97.34 | 85.24 |
| GPT4TS | [20] | 88.89 | 84.98 | 86.89 | 82.00 | 82.91 | 82.45 | 90.60 | 60.95 | 72.88 | 92.20 | 96.34 | 94.23 | 98.62 | 95.68 | 97.13 | 86.72 |
| **Peri-midFormer** | **(ours)** | 87.30 | 84.65 | 85.95 | 89.66 | 75.31 | 81.83 | 90.40 | 56.10 | 68.62 | 91.91 | 94.95 | 93.40 | 98.4 | 96.01 | 97.19 | 85.40 |

