# OpenReview forum: "Peri-midFormer: Periodic Pyramid Transformer for Time Series Analysis"
_NeurIPS.cc/2024/Conference — NeurIPS 2024 spotlight_

### Official Review · Reviewer_ECSx · 2024-07-05

**Soundness:** 3
**Presentation:** 2
**Contribution:** 2
**Rating:** 6
**Confidence:** 3

**Summary:**

This paper introduces Peri-midFormer to capture multi-periodicity in time series data. Specifically, it designs a pyramid structure and attention mechanisms to effectively model complex temporal variations. The proposed methods demonstrates great performance in several time series analysis tasks in the author's experiments.

**Strengths:**

1. This paper compares a variety of cutting-edge methods.
2. Overall, this paper is solid and the authors conducted thorough experiments.

**Weaknesses:**

None.

**Questions:**

1. Were the results of DLinear and PatchTST obtained with a lookback of 96? From the results in Table 12, it seems they were achieved with the default input of 336. Correspondingly, your assertion on line 236 is incorrect. As far as I know, DLinear, PatchTST, and FITS do not use a default input length of 96. FITS has a default lookback of 720.
2. Regarding the Anomaly Detection experiment, did you use point adjustment techniques and manual thresholds? Combining point adjustment techniques with very low manual thresholds (e.g., 0.5%) can easily achieve an F1 score greater than 90 on these datasets [1], even with a randomly generated anomaly score list. Please do not simply state that you followed previous work, as this only exacerbates the legacy issue. Clearly state your stance and clarify this issue appropriately in the paper.
3. Was the ablation study in Table 5 conducted only on the ETTh2 dataset? As far as I know, the ETTh2 dataset is small in scale, has relatively weak periodicity (compared to ETTh1, or multi-period datasets like Electricity and Traffic), and suffers from significant distribution drift. Under such circumstances, any hyperparameter adjustment or other random factors can greatly affect model performance.
4. The Exchange dataset belongs to the financial domain and typically exhibits weak periodicity (due to the unpredictability of financial data). This is evidenced by findings in the DLinear paper, where simply copying the last point can achieve SOTA performance. This contradicts two claims in your paper: (1) Peri-midFormer improves predictive accuracy by extracting periodicity, so how did it achieve SOTA performance on this weak-period dataset? (2) Your Limitation section mentions that Peri-midFormer is not good in scenarios with weak periodicity, yet your experimental results show that your method performs well in this scenario.

[1] Wagner, Dennis, et al. "Timesead: Benchmarking deep multivariate time-series anomaly detection." Transactions on Machine Learning Research (2023).

**Limitations:**

As mentioned above, the authors have already discussed the limitations, though I have some concerns for it.

---

> ### Author Rebuttal · Authors · 2024-08-03
>
> # Response to Reviewer ECSx
>
> Thank you for your detailed review and questions. Please find our answers below.
>
> ### **Q1: Were the results of DLinear and PatchTST obtained with a lookback of 96?**
>
> We have reviewed the results for DLinear and PatchTST in Table 12 and confirmed that you are correct—their look-back windows are not 96. We apologize for this mistake. Since most of Table 12's content was referenced from Table 14 of the GPT4TS paper, we assumed that the same look-back window was used across comparison methods.
>
> We also apologize for the incorrect assertion in line 236. Upon review, DLinear, PatchTST, and FITS indeed do not use the default look-back window length of 96. We have adjusted the experiments and used longer look-back windows (512 and 720) for these methods to ensure fairer comparisons. We have included the updated experiments in the second part of the global reply.
>
> ### **Q2: Regarding the Anomaly Detection experiment, did you use point adjustment techniques and manual thresholds?**
>
> You are correct that our brief statement about following the TimesNet approach was problematic. To address your question first, we did use point adjustment techniques with manual thresholding, and this framework was consistently applied across all comparison methods. Here is a detailed explanation of our anomaly detection strategy:
>
> **Training Phase:** During training, we applied a simple reconstruction loss to help the model learn the distribution of normal data.
>
> **Testing Phase:** For testing, we used the following code (showing only the main parts):
> ```python
> def test(self, setting):
>
>         attens_energy = []
>         self.anomaly_criterion = nn.MSELoss(reduce=False)
>
>         # (1) stastic on the train set
>         with torch.no_grad():
>             for i, (batch_x, batch_y) in enumerate(train_loader):
>                 outputs = self.model(batch_x, None, None, None)
>                 score = torch.mean(self.anomaly_criterion(batch_x, outputs), dim=-1)
>                 attens_energy.append(score)
>
>         attens_energy = np.concatenate(attens_energy, axis=0).reshape(-1)
>         train_energy = np.array(attens_energy)
>
>         # (2) find the threshold
>         attens_energy = []
>         test_labels = []
>         for i, (batch_x, batch_y) in enumerate(test_loader):
>             batch_x = batch_x.float().to(self.device)
>             outputs = self.model(batch_x, None, None, None)
>             score = torch.mean(self.anomaly_criterion(batch_x, outputs), dim=-1)
>             attens_energy.append(score)
>             test_labels.append(batch_y)
>
>         attens_energy = np.concatenate(attens_energy, axis=0).reshape(-1)
>         test_energy = np.array(attens_energy)
>         combined_energy = np.concatenate([train_energy, test_energy], axis=0)
>         threshold = np.percentile(combined_energy, 100 - self.args.anomaly_ratio)
>
>         # (3) evaluation on the test set
>         pred = (test_energy > threshold).astype(int)
>         test_labels = np.concatenate(test_labels, axis=0).reshape(-1)
>         gt = test_labels.astype(int)
>
>         # (4) detection adjustment
>         gt, pred = adjustment(gt, pred)
>
>         return
> ```
>
> The `adjustment` function is defined as:
>
> ```python
> def adjustment(gt, pred):
>     anomaly_state = False
>     for i in range(len(gt)):
>         if gt[i] == 1 and pred[i] == 1 and not anomaly_state:
>             anomaly_state = True
>             for j in range(i, 0, -1):
>                 if gt[j] == 0:
>                     break
>                 else:
>                     if pred[j] == 0:
>                         pred[j] = 1
>             for j in range(i, len(gt)):
>                 if gt[j] == 0:
>                     break
>                 else:
>                     if pred[j] == 0:
>                         pred[j] = 1
>         elif gt[i] == 0:
>             anomaly_state = False
>         if anomaly_state:
>             pred[i] = 1
>     return gt, pred
> ```
>
> We applied point adjustments using `gt, pred = adjustment(gt, pred)` to correct some false positives and false negatives.
>
> Additionally, we set a manual threshold with `threshold = np.percentile(combined_energy, 100 - self.args.anomaly_ratio)`, where `combined_energy` is the combined energy score of the training and testing sets, and it is a hyperparameter. The `anomaly_ratio` used for the different datasets are shown in the table below:
>
> | Datasets        | SMD  | MSL | SMAP | SWaT | PSM |
> |:---------------:|:----:|:---:|:----:|:----:|:---:|
> | anomaly_ratio   | 0.5  | 1   | 1    | 1    | 1   |
>
> As shown in the table, we did not apply excessive manual intervention for specific datasets.
>
> Of course, this information should have been included in the original paper, which is a shortcoming of our work. We will add this detailed explanation in the revised version.
>
> ### **Q3: Was the ablation study in Table 5 conducted only on the ETTh2 dataset?**
>
> Following your suggestions, we have added ablation experiments on several additional datasets, and we have included the results in the third part of the global response.
>
> ### **Q4: The Exchange dataset belongs to the financial domain and typically exhibits weak periodicity.**
>
> Our method does indeed maintain strong performance on some datasets with weaker periodicity, which may seem inconsistent with our focus. However, our approach involves more than just the periodic pyramid; it also incorporates time series decomposition. The success of our method on trend-dominant datasets like Exchange is largely due to this approach. Also as you mentioned the DLinear method, which also employs the operation of temporal decomposition, so in our experiments DLinear achieved very good performance on the Exchange dataset. We have provided a detailed response in the global reply, specifically in the fourth section.
>
> We had overlooked this phenomenon in the original paper, and your insight has helped us recognize it.
>
> Thank you once again for your valuable feedback. If you have any further questions, please feel free to ask.

---

> > ### Comment · Reviewer_ECSx · 2024-08-13
> >
> > Thanks for the response. It is necessary to include the new results and analysis in the revised paper. I have updated my rating.

---

> > > ### Author Response · Authors · 2024-08-13
> > >
> > > Thank you very much for recognizing our work, your suggestions make it more complete! We will add new experimental results, theoretical proofs, and further analysis to the revised version. Thank you for your time in reviewing the manuscript!

---

### Official Review · Reviewer_jAZw · 2024-07-10

**Soundness:** 3
**Presentation:** 4
**Contribution:** 3
**Rating:** 6
**Confidence:** 4

**Summary:**

In this paper, the authors proposed a new method Peri-midFormer, which uses the multi-periodicity of time series and modeling the periodic part of a time series in a pyramid way. They further proposed an attention mechanism to use the neighborhood relation in the pyramid. Extensive experiments on different tasks show the effectiveness of the proposed method.
After reading the rebuttal, I raise my rating from 5 to 6.

**Strengths:**

- The idea of using attention in the pyramid structure seems to be novel.
- Extensive experiments are conducted for different tasks on benchmark datasets.
- The proposed model is light-weight.

**Weaknesses:**

- Some important related works are missing.
  - There are other related works utilizing the idea of modeling time series in multi-scale, e.g. [1] [2] [3].
- Experiments could be improved.
  - The authors should clearly state whether they reproduce the results for comparison methods or they copy the numbers from the paper.
  - In the original paper of PatchTST, the authors use a longer context window than 96. The authors are suggested to tune this parameter for all comparison methods (as many papers did) and provide the best results, rather than fix the context window.
  - The authors should provide the script for hyper-parameter tuning to improve the reproducibility of the paper.
  - The ablation study (Table 5) is only shown on one dataset (ETTh2). The authors are suggested to include results in more datasets.
  - The training and inference complexity and actual time could be analyzed, as well as memory efficiency.
[1] TimeMixer: Decomposable Multiscale Mixing for Time Series Forecasting. ICLR 2024.
[2] PERIODICITY DECOUPLING FRAMEWORK FOR LONGTERM SERIES FORECASTING. ICLR 2024.
[3] Disentangling Structured Components: Towards Adaptive, Interpretable and Scalable Time Series Forecasting. TKDE 2024.

**Questions:**

- Apparently, the proposed method focused on periodic part forecasting. However, it works pretty well on the non-periodic forecasting task (short-term forecasting). For instance, there is no periodic in the 'Yearly' dataset, the proposed method still performs quite well. Can authors explain the possible reasons?
- It is not clear why the proposed methods have a smaller number of parameters than PatchTST. Can the authors explain more?
- It seems that the trend part is not used for classification tasks (from Figure 7). Can the author explain more?

**Limitations:**

Focused only on periodic signal

---

> ### Author Rebuttal · Authors · 2024-08-03
>
> # Response to Reviewer jAZw
>
> Thanks for your valuable comments. We will explain your concerns point by point.
>
> ### **Q1: Some important related works are missing**
>
> Thank you for your reminder. We have carefully read the paper you provided and found them very helpful for improving the background of our paper. We will include these works in the related work section of the revised version.
>
> ### **Q2: Experiments could be improved**
>
> Thank you very much for your valuable feedback. Here are our additions or modifications:
>
> 1. **The authors should clearly state whether they reproduce the results for comparison methods or they copy the numbers from the paper.**
>
>     We apologize for the oversight in the original paper. We have reproduced some of the experimental results of the comparison methods, but due to time constraints, we also included results from the papers of these methods. Specifically, for the long-term forecasting task, we reproduced results for TSLANet and FITS, while other results were taken from previous papers:
>
>     (1) Time-LLM and iTransformer are from Table 13 of the TSLANet paper.
>
>     (2) GPT4TS, Dlinear, PatchTST, TimesNet, FEDformer, Autoformer, Stationary, ETSformer, LightTS, Informer, and Reformer are from Table 14 of the GPT4TS paper.
>
>     We will include these details in the revised version.
>
>
> 2. **The authors are suggested to tune this parameter for all comparison methods and provide the best results.**
>
>     We apologize for this oversight. To ensure fair comparison, we have adjusted the look-back window for other comparison methods as per your suggestion and re-conducted the experiments. We have included this information in the second part of the global reply.
>
> 3. **The authors should provide the script for hyper-parameter tuning to improve the reproducibility of the paper.**
>
>     Thank you for your valuable feedback. We will explain our hyperparameter tuning script with the following example and will release the script along with all the code to improve reproducibility:
>
>     For the long-term forecasting task on the Electricity dataset with a prediction length of 96:
>
>     ```
>     for d_model in 64 128 256 512 768
>     do
>     for layers in {1..5}
>     do
>     for top_k in {2..5}
>     do
>     for batch_size in 4 8 16 32 64
>     do
>     for learning_rate in 0.0001 0.0002 0.0005 0.001 0.002
>     do
>
>     python -u run.py \
>     --layers $layers \
>     --d_model $d_model \
>     --top_k $top_k \
>     --learning_rate $learning_rate \
>     --batch_size $batch_size \
>     --......
>
>     done
>     done
>     done
>     done
>     done
>     ```
>     Based on this script, we recorded results from multiple experiments and selected the best-performing hyperparameters.
>
> 4. **The ablation study is only shown on one dataset (ETTh2)**
>
>     Based on your suggestion, we have added more ablation experiments, and we have included this information in the third part of the global reply.
>
> 5. **The training and inference complexity and actual time could be analyzed, as well as memory efficiency.**
>
>     Thank you for your suggestion. We have added the experiments as you recommended,and show it in the first part of the global reply section.
>
>
> ### **Q3: There is no periodic in the 'Yearly' dataset, the proposed method still performs quite well**
>
> Thank you for your insightful comments. Our method does indeed maintain good performance on datasets with weaker periodicity, which might seem mismatched with the focus of our approach. However, due to our use of time series decomposition, our method can effectively capture future trends by predicting the trend part even in datasets like Yearly that do not have obvious periodicity but exhibit significant trends. This is illustrated by the "trend part" in Figure 7 of the original paper. We have provided a detailed response in the global reply, specifically in the fourth section.
>
> ### **Q4: It is not clear why the proposed methods have a smaller number of parameters than PatchTST**
>
> Thank you for your insightful comments. The main reasons for this phenomenon are:
>
> 1.	**Number of patches:** If we consider each periodic component in our method as a patch, our total number of patches is smaller than that of PatchTST. Specifically, PatchTST uses a fixed number of 64 patches (there are two versions, 64 and 42, and the 64 version was used in the experiments), while the number of patches in our method is variable. In the experiments shown in Figure 16 of the original paper, we set $k$ to 3 for both tasks, resulting in 40 and 32 patches after periodic decomposition, which is less than the 64 patches in PatchTST.
>
> 2.	**Model Size Variance:** Not all comparison methods use the same model size. We refer to what was done in the TimeaNet experiments and retain the hyperparameters of the models of the original method, thus maximizing the best possible performance of the individual models. In the experiments shown in Figure 16, the PatchTST model has a dimension of 512 and 3 layers, while our model has a dimension of 16 and 2 layers. This difference in model size is another reason for the smaller parameter count in our method compared to PatchTST.
>
> ### **Q5: It seems that the trend part is not used for classification tasks.**
>
> Thank you for your insightful comments. As you noted, the trend part is not used in classification tasks (without time series decomposition) but follows the path indicated by the red arrow. In classification tasks, reconstructing the original data is unnecessary, so the trend part is not extracted and added back. Additionally, the trend part is a crucial discriminative feature for classification data, so it should not be separated from the original data before feature extraction. We acknowledge that this aspect was not explained in detail in the original paper, and we will provide a more thorough explanation in the revised version.
>
> Thank you again for your valuable time and constructive feedback. If you have any further questions, please feel free to ask.

---

> > ### Comment · Reviewer_jAZw · 2024-08-09
> >
> > Thanks for the authors for the response in the rebuttal. My main concerns about experiments are addressed. Therefore, I would like to raise my rating from 5 to 6.

---

> > > ### Author Response · Authors · 2024-08-09
> > >
> > > Thank you very much for recognizing our work, your suggestions are very helpful in improving the quality of our work, and thank you for your valuable time in reviewing the manuscript!

---

### Official Review · Reviewer_GBMh · 2024-07-12

**Soundness:** 3
**Presentation:** 4
**Contribution:** 3
**Rating:** 7
**Confidence:** 4

**Summary:**

The paper introduces Peri-midFormer, a novel transformer-based architecture designed for time series analysis.
By leveraging the multi-periodicity inherent in time series data, the model constructs a Periodic Pyramid structure that decouples complex periodic variations into inclusion and overlap relationships among different periodic components.
The proposed method incorporates self-attention mechanisms to capture dependencies between these periodic components, achieving state-of-the-art performance across five mainstream time series tasks: short- and long-term forecasting, imputation, classification, and anomaly detection.

**Strengths:**

S1.  The concept of decoupling time series data into a Periodic Pyramid is a valid point.
This new representation seems to capture the multi-periodicity of time series effectively.

S2. The effectiveness of the proposed method is extensively verified on five tasks.

**Weaknesses:**

W1. While the Periodic Pyramid and self-attention mechanisms are well-explained, the model's complexity might pose challenges for practical implementation and scalability, especially for users with limited computational resources.
It would be useful if the authors could report in the main body of the paper, at least the main conclusion of the training and inference time for the proposed method, compared with existing baselines, across the five tasks.

W2. The improvements provided by Peri-midFormer seem to be insignificant when compared with Time-LLM and GPT4TS on forecasting, imputation, and anomaly detection tasks.
It would be better if the author could further discuss this in the main body of the paper (e.g., discuss the time-accuracy tradeoff as shown in Appendix E.4 Complexity Analysis).

**Questions:**

Q1 (cr. W1). Add in the main body of the paper a discussion about the training and inference time, as well as some critical results and main conclusions.

Q2 (cr. W2). Add in the main body of the paper a discussion about the significance of the improvements or about the time-accuracy tradeoff for Peri-midFormer against Time-LLM and GPT4TS on forecasting, imputation, and anomaly detection tasks.

**Limitations:**

L1. It is not clear how the proposed periodic pyramid can be effectively and efficiently integrated into multi-dimensional time series.

---

> ### Author Rebuttal · Authors · 2024-08-03
>
> # Response to Reviewer GBMh
>
> Thanks for your valuable comments. We will explain your concerns point by point.
>
> ### **Q1: (cr. W1). Add in the main body of the paper a discussion about the training and inference time, as well as some critical results and main conclusions.**
>
> Thank you very much for your valuable feedback. Concerns about computational complexity and scalability were also raised by other reviewers. Therefore, we have supplemented our experiments in this area, including the complexity, time, and memory efficiency of training and inference. Please refer to the first part of the global reply for details. We will include this content in the revised version to make the main text more complete.
>
> ### **Q2: (cr. W2). Add in the main body of the paper a discussion about the significance of the improvements or about the time-accuracy tradeoff for Peri-midFormer against Time-LLM and GPT4TS on forecasting, imputation, and anomaly detection tasks.**
>
> Thank you very much for your valuable feedback. Our method does indeed perform worse than Time-LLM and GPT4TS on some tasks. However, due to page limitations, we placed the complexity validation in the appendix, which may affect readers' understanding of the paper. Additionally, we did lack validation and discussion on the time-accuracy trade-off in the original paper. We have addressed this in the first part of the global reply section and will include it in the revised version.
>
> In addition, due to the limitation of the amount of content that can be displayed, we currently only supplemented our experiments on the long-term forecasting task, and if necessary, we will also supplement more experiments on the imputation and anomaly detection tasks.
>
> ### **Q3: L1. It is not clear how the proposed periodic pyramid can be effectively and efficiently integrated into multi-dimensional time series.**
>
> We apologize for not clearly explaining the operational mechanism of our method in the original paper. The figures only showed the operation for a single channel, and although we mentioned retaining the original channels in line 132, this did not effectively convey our approach. We should have emphasized our use of a channel-independent strategy and clarified that the figures only illustrate operations for one channel. This will be addressed in the revised version.
>
> Thank you again for your valuable feedback, which has helped make our work more complete. If you have any further questions, please feel free to ask.

---

> > ### Comment · Reviewer_GBMh · 2024-08-12
> >
> > Thanks to the authors for addressing my concerns; I remain positive about this work as earlier. Look forward to future versions with more detailed discussions and new results as proposed.

---

> > > ### Author Response · Authors · 2024-08-13
> > >
> > > Thank you for recognizing our work! In future versions, we will add all new experimental results, theoretical proofs, and further discussions to make our work more complete. Thank you again for your valuable time in reviewing our paper!

---

### Official Review · Reviewer_W65V · 2024-07-13

**Soundness:** 3
**Presentation:** 3
**Contribution:** 2
**Rating:** 6
**Confidence:** 5

**Summary:**

The abstract succinctly introduces the Peri-midFormer, a novel approach designed for time series analysis, acknowledging the challenges posed by the discrete nature of time series data and the complexity of capturing periodic variations directly. It proposes a method to address these challenges by decomposing complex periodic variations into hierarchical periodic components, termed the periodic pyramid. This approach leverages inclusion and overlap relationships among these components, mimicking the natural pyramid structure observed in time series data.

**Strengths:**

Innovative Approach: The concept of a periodic pyramid to model time series data is innovative and promises to address the limitations of traditional methods that struggle with capturing complex periodic patterns.

Hierarchical Representation: By representing time series as a pyramid with progressively shorter periodic components, the model potentially enhances the understanding of multi-scale temporal relationships.

Self-Attention Mechanism: Incorporating self-attention into the periodic pyramid allows capturing intricate relationships among periodic components, which is crucial for tasks like anomaly detection and forecasting.

**Weaknesses:**

Complexity and Scalability: The introduction of a hierarchical pyramid structure combined with self-attention could potentially introduce computational complexities and scalability issues, especially with larger datasets or real-time applications. Addressing these concerns in the paper would strengthen its practical utility.

State-of-the-Art Comparison: It is not explicitly stated whether Peri-midFormer achieves state-of-the-art (SOTA) performance when compared to strong baseline models. Without clear comparative results, it is difficult to ascertain if the proposed method truly represents an advancement over current leading approaches in time series analysis.

Interpretability: The abstract lacks interpretability regarding why the periodic pyramid structure exists in the applications and why it plays a critical role in improving forecasting. Providing a rationale or theoretical justification for the efficacy of the pyramid structure in capturing temporal patterns would enhance the understanding and acceptance of the proposed method.

**Questions:**

see weakness

**Limitations:**

see weakness

---

> ### Author Rebuttal · Authors · 2024-08-03
>
> # Response to Reviewer W65V
> Thank you for your detailed review and questions. Please find our answers below.
>
> ### **Q1: Complexity and Scalability**
>
> We have adopted the suggestions from you and two other reviewers to include additional experiments, covering the complexity, time, and memory efficiency of both training and inference, to further validate the method's complexity and scalability. Please refer to the first part of the global reply section for details.
>
> ### **Q2: State-of-the-Art Comparison**
>
> In the original paper, we stated that our method achieved SOTA results (see line 77). Additionally, we compared the performance of various methods in the explanation of each experiment's results, highlighting the outstanding performance of our method. However, this may still be insufficient and could require more thorough discussion of the comparative results. If possible, we will provide a more comprehensive analysis of the comparative results in the revised version.
>
> ### **Q3: Interpretability**
>
> The abstract indeed lacks an explanation of the periodic pyramid structure in time series, which affects the understanding of the method. We will include this content in the revised version.
>
> Additionally, the lack of a clear explanation of the fundamental principles of the periodic pyramid makes it seem unsupported. Therefore, we have re-examined our method and analyzed its fundamental principles as follows:
>
> To demonstrate the essence of attention computation among multi-level periodic components, we need to analyze how the interactions between periodic components at different levels affect the final feature extraction.
>
> In time series analysis, different periodic components correspond to different time scales. This means that through decomposition, we can capture components of various frequencies within the time series. The essence of the periodic pyramid is to capture these different frequency components through its hierarchical structure.
>
> Using single-channel data as an example, and given that we adopt an independent channel strategy, this can be easily extended to all channels. Assume the time series $x(t)$ can be decomposed into multiple periodic components ${x_n}(t)$ :
>
> $$x(t) = \sum\limits_{n = 1}^N {{x_n}} (t) \tag{1}$$
>
> Taking two different periodic components as examples:
>
> $${x_i}(t) = {A_i}\sin \left( {\frac{{2\pi t}}{{{T_i}}} + {\phi _i}} \right),{x_j}(t) = {A_j}\cos \left( {\frac{{2\pi t}}{{{T_j}}} + {\phi _j}} \right) \tag{2}$$
>
> where $A$ is amplitude, $T$ is period, and $\phi $ is phase. Due to the overlap and inclusion relationships between different periodic components, we employ an attention mechanism in the periodic pyramid to capture the similarities between different periodic components, focusing on important periodic features. When applying the attention mechanism, we have:
>
> $${Q_i} = {W_Q}{x_i}(t),\quad {K_j} = {W_K}{x_j}(t),\quad {V_j} = {W_V}{x_j}(t) \tag{3}$$
>
> where ${W_Q}$ 、${W_K}$ and ${W_V}$ are learnable weight matrices. From equations (2) and (3):
>
> $${Q_i} = {W_Q}{A_i}\sin \left( {\frac{{2\pi t}}{{{T_i}}} + {\phi _i}} \right),\quad {K_j} = {W_K}{A_j}\cos \left( {\frac{{2\pi t}}{{{T_j}}} + {\phi _j}} \right) \tag{4}$$
>
> Further, the dot-product attention can be expressed as:
>
> $${Q_i}K_j^T = {A_i}{A_j}\left( {{W_Q}\sin \left( {\frac{{2\pi t}}{{{T_i}}} + {\phi _i}} \right)} \right){\left( {{W_K}\cos \left( {\frac{{2\pi t}}{{{T_j}}} + {\phi _j}} \right)} \right)^T} \tag{5}$$
>
> Using the trigonometric identity $\sin (a)\cos (b) = \frac{1}{2}[\sin (a + b) + \sin (a - b)]$, the dot-product ${Q_i}K_j^T$ can be further expressed as:
>
> $${Q_i}K_j^T = \frac{1}{2}{A _i}{A _j}\left\\{ {{W _Q}\left[ {\sin \left( {\frac{{2\pi t}}{{{T _i}}} + {\phi  _i} + \frac{{2\pi t}}{{{T _j}}} + {\phi _j}} \right) + \sin \left( {\frac{{2\pi t}}{{{T _i}}} + {\phi _i} - \frac{{2\pi t}}{{{T_j}}} - {\phi _j}} \right)} \right]} \right\\}{\left( {{W_K}} \right)^T} \tag{6}$$
>
> Based on this, considering the periodicity and symmetry of $\sin (a + b)$ and $\sin (a - b)$, when the periods of two time series components are close / same (**intra-level attention in the pyramid**, see the right side of Figure 3 in the original paper) or have overlapping / inclusive parts (**inter-level attention in the pyramid**, see the right side of Figure 3 in the original paper), the values of these two sine functions will be highly correlated, resulting in a large ${Q_i}K_j^T$ value. This indicates that the periodic pyramid model can effectively capture similar periodic patterns across different time scales.
>
> Next, incorporating this into the calculation of the attention score:
>
> $${s _{ij}} = \frac{{\exp \left( {\frac{{{Q _i}K _j^T}}{{\sqrt {{d _k}} }}} \right)}}{{\sum\limits _{m} {\exp } \left( {\frac{{{Q _i}K _{m}^T}}{{\sqrt {{d _k}} }}} \right)}} \tag{9}$$
> where $m$ denotes the index of all key values, including $j$. It can be seen that the attention scores between highly correlated periodic components will be higher, which we have already validated in Figures 13 and 14 of the original paper.
>
> From the above derivation, it can be seen that the attention mechanism can measure the similarity between different periodic components. This similarity reflects the alignment between different periodic components in the time series, allowing the model to capture important periodic patterns. By capturing these periodic patterns, the periodic pyramid can extract key features of the time series, resulting in a comprehensive and accurate time series representation. This representation not only includes information across different time scales but also enhances the representation of important periodic patterns.
>
> Due to character limitations, there are parts of the derivation will be explained inside the official comment.
>
> The above proof partially explains the effectiveness of the periodic pyramid feature extraction method.
> We hope this addresses your concerns. If you have any further questions, please feel free to ask.

---

> > ### Comment · Reviewer_W65V · 2024-08-13
> >
> > Thank the authors for the detailed reponse. I have raised my rating accordingly.

---

> > > ### Author Response · Authors · 2024-08-13
> > >
> > > Thank you very much for recognizing our work! Your review comments have made it more complete, especially with the additions to the theoretical proof. We will incorporate these improvements into the revised version. Thank you for taking the time to review our manuscript!

---

> ### Author Response · Authors · 2024-08-03
> **Additions to replies**
>
> # Additions to the answer to the third question (Q3: Interpretability):
> ### **1. Qualitative analysis of the essence of the periodic pyramid:**
>
> Qualitatively analyzing the essence of feature extraction through the periodic pyramid lies in the combination of its multi-level structure and attention mechanism:
>
> (1) Multi-level Structure: By decomposing the time series into multiple periodic components, the periodic pyramid can capture features at different time scales. This decomposition allows the model to handle both short-term and long-term dependencies at various levels.
>
> (2) Attention Mechanism: The attention mechanism can adaptively focus on the most relevant parts of the periodic pyramid, enhancing the model's focus on important features.
>
> (3) Feature Aggregation: By aggregating features from different levels, the periodic pyramid can generate a comprehensive representation of the time series that includes information from all periodic components. This aggregation ensures that the model can fully capture the complex dynamic patterns in the time series.
>
>
>
>
> ### **2. Additional proof of continuation of Equation (9):**
>
> Further, the attention vector ${{\bf{a}}_i}$ of ${x_i}(t)$ can be obtained as:
>
> $${{\bf{a}}_i} = \sum\limits_m {{s _{im}}} {V_m} \tag{10}$$
>
> where ${V_m} = {W_V}{x_m}(t) = {W_V}{A_m}\cos \left( {\frac{{2\pi t}}{{{T_m}}} + {\phi _m}} \right)$, therefore ${{\bf{a}}_i}$ can be expressed as:
>
> $${{\bf{a}} _i} = \sum\limits_m {\frac{{\exp \left( {\left\\{ {{W _Q}\left[ {\sin \left( {\frac{{2\pi t}}{{{T _i}}} + \frac{{2\pi t}}{{{T _m}}} + {\phi _i} + {\phi _m}} \right) + \sin \left( {\frac{{2\pi t}}{{{T _i}}} - \frac{{2\pi t}}{{{T _m}}} + {\phi _i} - {\phi _m}} \right)} \right]} \right\\}{{\left( {{W _K}} \right)}^T}/\sqrt {{d _k}} } \right)}}{{\sum\limits _{m} {\exp } \left( {\left\\{ {{W_Q}\left[ {\sin \left( {\frac{{2\pi t}}{{{T _i}}} + \frac{{2\pi t}}{{{T _{m}}}} + {\phi _i} + {\phi _{m}}} \right) + \sin \left( {\frac{{2\pi t}}{{{T _i}}} - \frac{{2\pi t}}{{{T _{m}}}} + {\phi _i} - {\phi _{m}}} \right)} \right]} \right\\}{{\left( {{W _K}} \right)}^T}/\sqrt {{d _k}} } \right)}}} {W _V}{A _m}\cos \left( {\frac{{2\pi t}}{{{T _m}}} + {\phi  _m}} \right) \tag{11}$$
> where $m$ is the same as in Equation (7) in the original paper and is used for selecting components that have interconnected relationships with ${x_i}(t)$. Equation (11) is the expanded form of Equation (7) in the original paper, which leads to an explanation for the good performance of the Periodic Pyramid Attention Mechanism in capturing the periodic properties of the different levels in the time series.

---

### Author Rebuttal · Authors · 2024-08-03

# General Responses
We thank the Reviewers for the insightful comments and detailed feedback. Here's the global reply.

### **1. Validation of Computational Complexity and Scalability**

Following the suggestions of several reviewers, we have supplemented our experiments with tests on computational complexity and scalability, specifically including training and inference complexity, actual time, and memory usage. We conducted these experiments on larger datasets (Electricity and ETTh1) compared to the ETTh2 dataset used in the original paper. The results are presented in Tables 1 and 2 (Due to the long training time of Time-LLM, the relevant metrics for its inference on the Electricity dataset have not been collected yet).

As shown, our proposed Peri-midFormer demonstrates a significant advantage in computational complexity on the Electricity dataset, without the excessive inference time concerns raised by several reviewers, and achieves the lowest MSE. Similarly, on the ETTh1 dataset, the computational overhead and inference time of Peri-midFormer do not pose a disadvantage. In fact, while achieving an MSE second only to Time-LLM, Peri-midFormer’s computational overhead and inference time are substantially lower than those of Time-LLM.

This analysis demonstrates that our method has notable advantages in terms of computational complexity and scalability.

### **2. Adjustment of Lookback Window for Comparison Methods**

Due to our oversight, the original paper inaccurately described the lookback window for some comparison methods and used inappropriate lookback windows for others, leading to less rigorous experiments. Based on suggestions from several reviewers, we have adjusted the lookback windows for some comparison methods. Specifically, we have set the lookback windows for FITS, DLinear, PatchTST, TimesNet, and Pyraformer to 512, consistent with our Peri-midFormer. Additionally, since FITS originally had a lookback window of 720, we have included experiments with a 720 look-back window for comparison with Peri-midFormer. The results are presented in Table 3.

As seen from the table, several comparison methods benefit from the extended look-back window, showing some improvement in prediction performance compared to the results in Table 12 of the original paper. However, there is still a noticeable gap compared to our Peri-midFormer. Furthermore, in the comparison with a 720 look-back window, FITS does not perform as well as our Peri-midFormer.
This adjustment ensures a fairer experiment and highlights the advantages of Peri-midFormer.

### **3. Ablation Study Adjustment**

In the original paper, our ablation study was only validated on the ETTh2 dataset. Based on suggestions from several reviewers, we have expanded the experiments to include more datasets. We have supplemented the ablation study with experiments on the ETTh1, Electricity, Weather, and Traffic datasets, as shown in Table 4. It can be seen that each module we proposed performs effectively across multiple datasets, further demonstrating the superior performance of Peri-midFormer.


### **4. Explanation for Peri-midFormer's Strong Performance on Non-periodic Datasets**

In the original paper, we mentioned that Peri-midFormer excels on datasets with strong periodicity, and performs poorly on those with weak periodicity. However, as pointed out by several reviewers, Peri-midFormer has shown outstanding performance on the Exchange dataset in long-term forecasting tasks, and on the Yearly dataset in short-term forecasting tasks, both of which lack clear periodicity. This contradicts our initial description.
Upon careful examination of the Exchange and Yearly datasets, we found that while they indeed lack obvious periodicity, they exhibit strong trends, as shown in Figure 1-6. This explains why Peri-midFormer performs well on these datasets. Peri-midFormer employs a temporal decomposition strategy, which involves separating the trend part from the original data before partitioning the periodic components. The trend part is then added back after the output of Peri-midFormer, as illustrated in Figure 7 of the original paper.
To be candid, we adopted temporal decomposition to mitigate the influence of the original data's periodic characteristics, thereby enhancing Peri-midFormer's effectiveness. This approach was repeatedly explained in the original paper. Thanks to temporal decomposition, Peri-midFormer can leverage trend prediction to achieve excellent performance on datasets like Exchange and Yearly, which exhibit strong trends.

We sincerely thank the reviewers for their valuable suggestions, which have made our work more comprehensive. If possible, we will incorporate all supplementary content in the revised version. Once again, we appreciate the reviewers' time and insightful feedback!

---

### Author Response · Authors · 2024-08-12

Dear Reviewers, we have responded to your questions in detail. If you have any additional concerns, please let us know, and we will do our best to further refine our work. Thank you again for your valuable time!

---

### Decision · Program_Chairs · 2024-09-25

**Decision:**

Accept (spotlight)

**Comment:**

There is a clear consensus in the PC that this is a strong paper that should be accepted. Congratulations to the authors! I am requesting the author to carefully incorporate the suggestion provided by the review in the camera-ready version.